



# Sea salt reactivity over the northwest Atlantic: An in-depth look using the airborne ACTIVATE dataset

Eva-Lou Edwards[1], Yonghoon Choi[2,3], Ewan C. Crosbie[2,3], Joshua P. DiGangi[2], Glenn S. Diskin[2], Claire E. Robinson[2,3,†], Michael A. Shook[2], Edward L. Winstead[2,3], Luke D. Ziemba[2], and Armin Sorooshian[1,4]

[1]Department of Chemical and Environmental Engineering, University of Arizona, Tucson, AZ, 85721, USA
[2]NASA Langley Research Center, Hampton, VA, 23681, USA
[3]Analytical Mechanics Associates, Inc., Hampton, VA, 23666, USA
[4]Department of Hydrology and Atmospheric Sciences, University of Arizona, Tucson, AZ, 85721, USA

[†]Deceased

[*]Corresponding author: armin@arizona.edu





## Abstract

Chloride (Cl⁻) displacement from sea salt particles is an extensively studied phenomenon with implications on human health, visibility, and the global radiation budget. Past works have investigated Cl⁻ depletion over the northwest Atlantic (NWA), yet an updated, multiseasonal, and geographically expanded account of sea salt reactivity over the region is needed. This study uses chemically resolved mass concentrations and meteorological data from the airborne Aerosol Cloud meTeorology Interactions oVer the western ATlantic Experiment (ACTIVATE) to quantify seasonal, spatial, and meteorological trends in Cl⁻ depletion and to explore the importance of quantifying (1) non-sea salt sources of $Na^+$ and (2) mass concentrations of lost Cl⁻ instead of relative amounts displaced. Lost Cl⁻ mass concentrations are lowest in December-February and March, moderate around Bermuda in June, and highest in May (median losses of 0.04, 0.04, 0.66, and 1.76 µg m⁻³, respectively), with losses in May high enough to potentially accelerate tropospheric oxidation rates. Inorganic acidic species can account for all Cl⁻ depletion in December-February, March, and June near Bermuda, yet none of the lost Cl⁻ in May, suggesting organic acids may be of importance for Cl⁻ displacement in certain months. Contributions of dust to $Na^+$ are not important seasonally but may cause relevant overestimates of lost Cl⁻ in smoke and dust plumes. Higher percentages of Cl⁻ depletion often do not correspond to larger mass concentrations of lost Cl⁻, so it is highly recommended to quantify the latter to place depletion reactions in context with their role in atmospheric oxidation and radiative forcing.





## 1. Introduction

Chlorine (Cl) is a common constituent of trace gases and aerosol particles found in Earth's atmosphere. Chlorine-containing species play a critical role in the global radiation budget for many reasons, including their ability to produce highly reactive Cl radicals. These radicals can perturb atmospheric chemical processes by inducing reactions that would otherwise be less likely to occur and/or accelerating the rates of certain reactions. For example, Cl radicals in the stratosphere can incite reactions that destroy ozone ($O_3$; Molina and Rowland, 1974; Solomon et al., 2023), therefore allowing increased amounts of shortwave radiation to reach the surface and harmfully affect living beings.

Cl radicals typically react faster with volatile organic compounds (VOCs) compared to hydroxyl radicals (OH; Roberts et al., 2008; Thornton et al., 2010; Young et al., 2014), which has particular importance in the troposphere. Cl radicals oxidize methane ~16 times faster than OH (Faxon and Allen, 2013 and references therein), thus reducing the lifetime of this important greenhouse gas. Accelerated oxidation of methane and other VOCs can result in increased $O_3$ production near the surface (Knipping and Dabdub, 2003; Pechtl and von Glasow, 2007; Tanaka et al., 2003), which can have deleterious effects on animals ( e.g., respiratory problems, increased mortality; Lippmann, 1989; Nuvolone et al., 2018) and plants (e.g., decreased growth and photosynthesis; Wittig et al., 2009). Cl radicals may be responsible for 15 – 27% of VOC oxidation in the global troposphere (Sherwen et al., 2016) and can play an exceptionally critical role in governing atmospheric composition in the early morning when OH radicals are less abundant (Young et al., 2013; Riedel et al., 2014; Osthoff et al., 2008). Due to their significant impacts on radiative forcing, rates of chemical cycling, and the health of living organisms, it is critical to quantify and understand sources of atmospheric Cl radicals.

Sea salt aerosol particles are the largest reservoir of reactive atmospheric Cl. Keene et al. (1999) estimates that at any given time there are ~22 Tg of reactive Cl in the troposphere, and that 68% of this mass is found in particulate form, primarily sea salt. Although the Cl in sea salt will not directly photolyze to produce Cl radicals, it can be displaced by acidic species (e.g., sulfate [$SO_4^{2-}$], nitrate [$NO_3^-$], organic acids) and released in a reactive gaseous form (e.g., $ClNO_2$, HCl, $Cl_2$) that has the potential to produce Cl radicals. This phenomenon is called chloride ($Cl^-$) depletion and can be generalized with the following reaction:

$$HA + NaCl \rightarrow NaA + HCl_{(g)} \qquad (R1)$$

where A is one of the acidic species mentioned above. In addition to producing reactive chlorine-containing gases, $Cl^-$ depletion can alter the acidity (e.g., Keene and Savoie, 1998), hygroscopicity (e.g., Drozd et al., 2014; Ghorai et al., 2014; Randles et al., 2004), and optical properties (Finlayson-Pitts and Pitts, 2000; Tang et al., 1997) of sea salt particles. Such changes affect partitioning of other chemicals (e.g., water vapor, ammonia [$NH_3^+$], $SO_4^{2-}$, $NO_3^-$) between the gas and particle phases (Chen et al., 2021), the rates and types of reactions occurring within sea salt particles (Chameides and Stelson, 1993), the activity of these particles as cloud condensation nuclei (e.g., Chatterjee et al., 2020), and their interactions with solar radiation, all of which can have implications for visibility, air quality, biogeochemical cycles, and Earth's radiation budget.

Many factors dictate the extent to which $Cl^-$ depletion occurs in an air mass including meteorology (e.g., wind speed, temperature, relative humidity [RH], available solar radiation), the size distribution and mixing state of sea salt particles, and the availability and length of exposure to surrounding acidic species (Su et al., 2022 and references therein). Regarding the latter, $Cl^-$



depletion is therefore typically observed where marine particles and acidic species are both
present, such as where emissions from biomass burning (BB) advect over a marine location (Braun
et al., 2017; Maudlin et al., 2015; Li et al., 2003; Yokelson et al., 2009; Akagi et al., 2013; Dang
et al., 2022; Crosbie et al., 2022), in regions with active phytoplankton and marine bacteria that
emit dimethyl sulfide (DMS), which can oxidize to form sulfuric acid ($H_2SO_4$; Seinfeld and Pandis,
2016; Tang et al., 2019; Yan et al., 2020), and/or in and around urban coastal environments (e.g.,
Kong et al., 2014; Chatterjee et al., 2020; AzadiAghdam et al., 2019; Nolte et al., 2008) where
anthropogenic emissions serve as precursors for various acidic species.
For this reason, the northwest Atlantic (NWA) is an opportune region for observing and
studying $Cl^-$ depletion. Cities extending along the East Coast of North America consistently emit
sulfur dioxide ($SO_2$), nitrogen oxides ($NO_x$), and VOCs, which can oxidize to form $H_2SO_4$, nitric
acid ($HNO_3$), and organic acids, respectively, while sea salt particles are ubiquitous over the region
due to wave breaking (Reid et al., 2001; Ferrare et al., 2023). Occasional long-range transport from
BB in Alaska, Canada, and the western United States (U.S.; Fehsenfeld et al., 2006; Mardi et al.,
2021), agricultural fires throughout the eastern and southeastern U.S. (Jaffe et al., 2020; McCarty
et al., 2007), wintertime wood burning for residential heating (Corral et al., 2021; Sullivan et al.,
2019), and seasonally varying emissions from vegetation and ocean biological activity (Savoie et
al., 2002; Corral et al., 2022) can also introduce acidic species to this region.
$Cl^-$ depletion has been observed over the NWA for decades (Table 1). Previous datasets
typically span 2 – 3 months, and most are reflective of conditions during the boreal summer,
although there are a handful of studies extending outside of this period (i.e., Keene et al., 1990;
Yao and Zhang, 2012; Zhao and Gao, 2008; Haskins et al., 2018). Combining results from these
works to build seasonal and temporal statistics is challenged by the fact that each dataset is specific
to a certain altitude (or range of altitudes), location(s), time period, sampling method, and size
range of sampled particles. In addition to these logistical constraints, there is an overall shortage
of $Cl^-$ depletion data for the spring, fall and winter, which is of concern as depletion processes are
sensitive to several properties that fluctuate seasonally over the NWA (e.g., temperature, solar
radiation, RH).
Most past works over the NWA report on $Cl^-$ depletion along the United States East Coast
(USEC) and/or at Bermuda. To our knowledge, there is an absence of discussion about the gradient
in $Cl^-$ depletion moving from the USEC to the open ocean environment closer to Bermuda. Corral
et al. (2021) showed strong gradients in aerosol optical depth along this direction for several
particle types including sea salt and $SO_4^{2-}$, suggesting there may be a gradient in $Cl^-$ depletion as
well. Furthermore, $Cl^-$ depletion results from previous studies typically reflect conditions near the
surface, yet Shinozuka et al. (2004) showed that the vertical scattering profile of sea salt in the
lower 1 km of the atmosphere becomes increasingly less uniform with increasing wind speed. Also
of note is that most datasets referenced in Table 1 are now several decades old. Mass
concentrations of $SO_2$ , $NO_x$ , $SO_4^{2-}$, and $NO_3^-$ over the eastern U.S. and Canada have steadily
decreased since 1990 due, in part, to the Clean Air Act of 1963 and its subsequent amendments
(Feng et al., 2020; Kuklinska et al., 2015). Such reductions warrant an updated analysis of $Cl^-$
depletion over the NWA.
We note that $Cl^-$ depletion results from the Wintertime Investigation of Transport, Emissions
and Reactivity (WINTER) aircraft campaign (Haskins et al., 2018) are an exception to many of
the points raised above. As an airborne campaign from February – March 2015, WINTER provides
data relevant to halogen chemistry at altitudes throughout the boundary layer, at a time of year that
had previously not been studied, and in a year recent enough to capture the aforementioned



reductions in anthropogenically sourced acidic species. However, WINTER flights specifically
sampled over and downwind of various pollution sources in the eastern and southeastern U.S.,
meaning Cl⁻ depletion results may be disproportionately reflective of highly polluted, coastally
influenced air masses as compared to other air mass types observed over the NWA during winter
and spring (e.g., those (i) occurring after synoptically forced frontal systems have moved through,
(ii) associated with cold air outbreaks (CAOs), and (iii) occurring when southerly winds advect
maritime air masses northward along the East Coast).
137        It is common for Cl⁻ depletion studies to base their calculations on the assumption that sea salt
particles are the only source of atmospheric sodium ($Na^+$; i.e., $Na^+$ is used as the reference species
for determining the extent of Cl⁻ depletion observed), including nearly all the works listed in Table
1. The validity of this assumption is dependent on several factors, including the proximity to urban
emissions, if dust particles are present, and the size range of particles sampled. Ooki et al. (2002)
found $Na^+$ to be highly correlated with potassium ($K^+$) in particles < 1.1 µm in urban air masses,
implying that these two species have the same source in fine, anthropogenically sourced particles.
$K^+$ is thought to come mainly from BB (Echalar et al., 1995; Andreae et al., 1998; Andreae and
Merlet, 2001) and anthropogenic activities (Ooki et al., 2002 and references therein), suggesting
that marine air masses heavily influenced by BB or urban emissions may have nonnegligible
contributions from non-sea salt sources to total $Na^+$, especially if submicron particles contribute
significantly to total mass concentrations (which would depend on the size range of particles
sampled). $Na^+$ can also be found in mineral dust (Seinfeld and Pandis, 2016), which has motivated
a handful of studies to discern between the amounts of $Na^+$ coming from dust and sea salt using a
system of equations (e.g., Boreddy and Kawamura, 2015; AzadiAghdam et al., 2019). The NWA
is known to be periodically influenced by Asian, African, and North American dust (e.g., Aldhaif
et al., 2020) and emissions from BB (Fehsenfeld et al., 2006; Schroder et al., 2018; Sullivan et al.,
2019; Mardi et al., 2021), and is consistently influenced by anthropogenic activities throughout
the year. Several works shown in Table 1 have acknowledged that these additional sources of $Na^+$
may influence estimates of Cl⁻ depletion over the NWA, but none have quantitatively explored this
possibility.
158        Finally, most Cl⁻ depletion studies report the percentage of Cl⁻ in unreacted sea salt particles
that has been displaced by acidic species, an approach useful for quantifying the extent of Cl⁻
depletion processes independently of the sea salt mass concentrations present, which can vary
seasonally, temporally, and geographically. However, reporting Cl⁻ depletion as a percentage can
make it more difficult to conceptualize and quantify the degree to which depletion reactions may
be affecting atmospheric oxidation potential. Several past works focusing on the NWA have
reported the magnitude of Cl⁻ displaced from sea salt particles, either in units of nmol m⁻³ (e.g.,
Keene and Savoie, 1998; Keene et al., 1990) or pptv (Keene et al., 2007; Haskins et al., 2018),
which we find useful for comprehensive interpretation considering that Singh and Kasting (1988)
suggested ppbv concentrations of gaseous and reactive Cl species (e.g., HCl) have the potential to
produce enough Cl radicals to oxidize 20 – 40% of nonmethane alkanes in the marine troposphere.
Thus, reporting Cl⁻ depletion both as a percentage and as a mass concentration benefits the
atmospheric chemistry community as results can be used either comparatively or to improve
quantification of Cl radical budgets and the atmospheric oxidation capacity in a given region.
Although a few past works in the NWA have reported mass concentrations of displaced Cl⁻, there
is still a need for results reflecting current conditions across a range of seasons as we have
discussed above.





In summary, there is a demand for an updated, multi-seasonal, spatially resolved dataset
reflecting $Cl^-$ depletion processes in the NWA boundary layer across a variety of meteorological
conditions and air mass types. There is also interest in (i) exploring the sensitivity of $Cl^-$ depletion
results to accounting for non-sea salt sources of $Na^+$, especially in seasons and/or air masses
influenced by dust and BB emissions, as well as (ii) quantifying both the percentage and magnitude
of $Cl^-$ displaced from sea salt particles for straightforward comparisons to other works and to link
results more easily to boundary layer Cl radical budgets and their potential influence on
atmospheric oxidation rates. This study seeks to address these points by using data from the NASA
Aerosol Cloud meTeorology Interactions oVer the western ATlantic Experiment (ACTIVATE)
airborne field campaign (Sorooshian et al., 2019). The statistical approach, large number of flights
spanning a range of seasons and meteorological conditions, and type of instruments deployed on
this campaign make the ACTIVATE dataset well-suited to address several of the outstanding
uncertainties and unknowns regarding $Cl^-$ depletion over the NWA.





**Table 1.** Relevant information from previous works, sorted chronologically, documenting Cl⁻
depletion over the Northwest Atlantic (NWA). "USEC" stands for United States East Coast, and
"U.S." stands for United States.

| Reference(s) | Dates | Location | Platform(s) | Reference species to determine Cl⁻ depletion | Discusses possibility of non-sea salt sources of Na⁺ and/or Cl⁻ |
|---|---|---|---|---|---|
| Keene et al. (1990) | Jul – Sep 1988 | USEC and near Bermuda | Ship and aircraft | $Na^+$ | No |
| Keene and Savoie (1998) | Apr – May 1996 | Bermuda | Surface station | $Na^+$ | No |
| Nolte et al. (2008) | May – Jun 2002 | Tampa, Florida (U.S.) | Surface stations | $Na^+$ | Yes |
| Yao and Zhang (2012) | Jun – Jul 2002, Oct – Nov 2002 | Kejimkujik, Nova Scotia | Surface station | $Na^+$ | No |
| Keene et al. (2004) | Jul – Aug 2002 | USEC | Ship | $Mg^{2+,\,2}$ | No |
| Quinn and Bates (2005) | Jul – Aug 2002 | USEC | Ship | $Na^+$ | No |
| Keene et al. (2007) | Jul – Aug 2004 | Appledore Island, Maine (U.S.) | Surface station | $Na^+$ and $Mg^{2+}$ | Yes |
| Zhao and Gao (2008) | Jul – Sep 2006 | Newark, New Jersey (U.S.) | Surface station | $Na^+$ | Yes |
| Bondy et al. (2017) | Jun – Jul 2011 | Centreville, Alabama (U.S.) | Surface station | $Na^+$ and $Mg^{2+}$ | Yes |
| Haskins et al. (2018) | Feb – Mar 2015 | USEC and over land around major pollution sources across the eastern U.S.[1] | Aircraft | $Na^+$ | Yes |

[1]The Wintertime Investigation of Transport, Emissions, and Reactivity (WINTER) airborne field
campaign focused on three regions over the U.S.: i) the northeast metropolitan corridor
(encompassing major cities from Boston to Washington D.C.), ii) the Ohio River Valley, and iii)
the Southeast. Research flights also extended over coastal waters to sample polluted air masses
downwind from their sources.
[2]Magnesium ($Mg^{2+}$) was chosen as the reference species for sea salt in Keene et al. (2004) as $Na^+$
had a relatively higher and more variable background in the quartz-fiber sampling media used.



## 2. Data and methods

### 2.1 ACTIVATE campaign description

The ACTIVATE field campaign focused on characterizing relationships between aerosol particles, meteorology, and marine boundary layer clouds over the NWA using two research aircraft flying in coordination. Operations were based out of NASA Langley Research Center (LaRC), although a multitude of other sites supported various aspects of the project. The high-flying King Air usually flew steadily at ~9 km releasing dropsondes and using a suite of remote sensors to retrieve particle and cloud properties below the aircraft. The low-flying HU-25 Falcon (hereafter referred to as the "Falcon") made in situ measurements of trace gases, aerosol particle properties, cloud and precipitation properties (if present), and meteorological conditions in and around boundary layer clouds or in clear conditions usually below 3 km.

ACTIVATE placed a high priority on building statistics to fulfill its objectives and address current uncertainties regarding aerosol-cloud interactions and remote sensing capabilities over the NWA. To acquire such statistics, the Falcon and King Air achieved 174 and 168 flights with 574 and 592 total flight hours, respectively, from 2020 – 2022 (note that 162 of these were "joint" flights where the aircraft flew in coordination; Sorooshian et al., 2023). The campaign included multiple seasons, with each aircraft adhering to an intentional and consistent flight strategy throughout, to better constrain the multitude of variables affecting a given clear or cloudy scene. As mentioned above, the King Air flew fixedly at ~9 km regardless of the amount of cloud coverage below. In the presence of low-level (<3 km) clouds, the Falcon conducted "cloud ensembles" by flying 3-minute legs at the following key vertical positions: near the ocean surface (MinAlt; ~150 m), below cloud base, above cloud base, below cloud top, and above cloud top. In the absence of low-level clouds, the Falcon switched to "clear ensembles," which involved 3-minute legs at MinAlt, ~230 m (an altitude useful for remote sensing validation), and at altitudes falling slightly below and above the boundary layer height (see Fig. 2 in Sorooshian et al. [2023] for an illustration of these ensembles). The campaign was executed over six deployments, which are referred to as Winter 2020 (February – March 2020), Summer 2020 (August – September 2020), Winter 2021 (January – April 2021), Summer 2021 (May – June 2021), Winter 2022 (November 2021 – March 2022), and Summer 2022 (May – June 2022) as recommended in Sorooshian et al. (2023). Note that Winter 2022 includes two months in 2021 but is referred to as "Winter 2022" for simplicity.

### 2.2 Falcon data

The main instrument providing data for this study is a particle into liquid sampler (PILS; Brechtel Manufacturing Inc. [BMI]) that was operated downstream from an isokinetic Clarke-style shrouded solid double-diffuser inlet (BMI; McNaughton et al., 2007) onboard the Falcon. The PILS grows aerosol particles with diameters of 50 - 5000 nm at ambient RH into droplets large enough to be collected via inertial impaction (Sorooshian et al., 2006; Crosbie et al., 2020). Droplets striking the impaction plate are pumped into vials that are analyzed offline using ion chromatography (IC) to quantify air equivalent mass concentrations of $Na^+$, ammonium ($NH_4^+$), $K^+$, magnesium ($Mg^{2+}$), calcium ($Ca^{2+}$), $Cl^-$, $NO_3^-$, $SO_4^{2-}$, and oxalate. PILS data are critical to this study due to the instrument's ability to capture particles containing sea salt, dust, and other refractory species that are largely omitted by the aerosol mass spectrometer (AMS). PILS flowrates were set such that it took 300 - 420 s (5 – 7 minutes) to fill each vial, the minimum duration for collecting enough particle mass to be above speciated detection limits while also meeting injection



volume requirements for IC analysis. Note that the time spent collecting one PILS sample is greater
than the duration of the individual level legs (~3 minutes) comprising clear and cloudy ensembles.
The possibility that each PILS sample could represent atmospheric properties sampled during
multiple level legs and/or periods of ascent or descent between level legs impacted our analysis in
two ways. First, PILS measurements must be considered as a representation of water-soluble ionic
composition throughout the lower 3 km of the atmosphere, meaning they cannot provide vertically
resolved information. Second, we exclude PILS data collected during cloudy ensembles to
eliminate possible cloud contamination. During cloudy ensembles, it is likely that the Falcon
intercepted a cloud within any interval of 5 – 7 minutes, and in doing so, shattered droplets and
other cloud artifacts were collected in the awaiting sample vial. Additionally, while flying through
clouds, large droplets and ice particles can impact onto the walls within the isokinetic inlet where
they may resuspend and, therefore, cause delayed sampling, of larger particles previously caught
on these walls.
The PILS was operated without upstream acid and base denuders, which opened the possibility
for soluble gases (e.g., $NH_3$) to contribute to speciated mass concentrations. During quality control
analyses, PILS $NH_4^+$ mass concentrations were unjustifiably high in many samples, prompting us
to omit this species from this study's analysis. As $NH_4^+$ is a critical species for deriving parameters
relevant to $Cl^-$ depletion, we alternatively use $NH_4^+$ mass concentrations from a high-resolution
time-of-flight aerosol mass spectrometer (HR-ToF-AMS; Aerodyne; DeCarlo et al., 2008;
hereafter referred to as an "AMS"), which provided non-refractory mass concentrations of $NH_4^+$
(among other species) for particles 60 – 600 nm in diameter at a 30-s time resolution. The AMS
additionally provided mass concentrations of spectral markers for organic components, of which
we use the tracers for oxygenated organics, *m/z* 44, and methanesulfonic acid (MSA), *m/z* 79. AMS
data were filtered to isolate those from clear ensembles and then averaged over the 5- to 7-minute
interval for each PILS sample. Due to differences in the size range of the PILS and AMS, $NH_4^+$
mass concentrations from the AMS represent a lower limit in this analysis.
Horizontal wind speed and static air temperature data were obtained using the Turbulent Air
Motion Measurement System (TAMMS; Thornhill et al., 2003) operating at 20 Hz time resolution,
while the diode laser hygrometer (DLH; Diskin et al., 2002) supplied water vapor mixing ratios
and values of RH at 1 Hz time resolution. A commercial cavity ringdown spectrometer (G2401-
m; PICARRO, Inc.) provided carbon monoxide (CO) measurements at 0.4 Hz resolution (DiGangi
et al., 2021), which are used to qualitatively compare the extent to which certain seasons were
influenced by anthropogenic emissions (Panagi et al., 2020; Naeher et al., 2001; Saide et al., 2011).
Data are only considered from clear ensembles for each of the parameters described in this
paragraph.
The Falcon occasionally intercepted clouds during clear ensembles. During these cloud passes,
certain instruments (e.g., the AMS) sampled downstream of a counterflow virtual impactor (CVI;
BMI; Shingler et al., 2012) for droplet residual characterization. We removed data collected during
periods with active CVI sampling from our analysis for all variables mentioned above.
**2.3 Deployment selection and season/category classifications**
This analysis focuses on data collected during the Winter 2022 and Summer 2022
deployments as they cover the largest geographical range over the NWA, thus presenting the best
opportunity for studying spatial gradients in $Cl^-$ depletion. During Winter 2022, sampling was
extended northward on flights when the Falcon flew to Quonset State Airport in Rhode Island,
refueled, and returned to LaRC, an option that was unavailable during the first four deployments





due to challenges associated with the COVID-19 pandemic. Summer 2022 is the only deployment
to (i) execute "transit flights" (i.e., flights where the Falcon flew to Bermuda, refueled, and flew
back to LaRC on the same day) and (ii) include a set of out-and-back flights based in Bermuda.
Additionally, Winter 2022 and Summer 2022 supply the largest and most continuous dataset
compared to the first two years of the campaign. Nearly half of the total Falcon flights occurred
within these two deployments, and sampling occurred consistently from 31 November 2021 to 18
June 2022 with a brief break from 30 March – 02 May 2022. The high frequency of flights over a
~7-month period allows us to explore the seasonal evolution of properties relevant to Cl$^-$ depletion,
while also observing their fluctuations on daily to multiday time scales.
299        To capture both seasonal and spatial trends, Winter 2022 and Summer 2022 data are distributed
among the following categories by season/month and/or by the geographical area sampled:
December-February (30 November 2021 – 26 February 2022), March (02 – 29 March 2022), May
(03 – 20 May 2022), March transit (22 March 2022), May transit (18, 21, and 31 May 2022), and
June Bermuda (02 – 13 June 2022). Note that some flights from the Winter 2022 and Summer
2022 deployments are omitted from this study because they are either composed entirely of cloudy
ensembles and/or PILS data are unavailable during the clear ensembles. To explore relationships
between (i) speciated mass concentrations and Cl$^-$ depletion, and (ii) phenomena occurring on finer
time scales (e.g., the passage of weather fronts, transport events of African dust plumes),
meteorological conditions and/or notable influence from distinct aerosol types are documented for
each research flight (RF). We also select RFs sampling various airstreams associated with passing
frontal systems and dust-influenced air masses to further illustrate relationships between these
phenomenon and properties relevant to Cl$^-$ depletion.

**2.4 Calculations relevant to Cl$^-$ depletion**
314        The following section describes how various properties associated with Cl$^-$ depletion were
derived using PILS and AMS bulk speciated mass concentrations and literature-based ratios for
ions in sea salt, dust, and emissions from various combustion processes. Identifying the amount of
Cl$^-$ displaced from sea salt particle begins with quantifying the original amount of Cl$^-$, which we
derive from Na$^+$ in sea salt (ssNa$^+$) as this species has a relatively high mass fraction and is
chemically inert in sea salt particles. We use Eqs. 1 – 5 to resolve contributions of sea salt and dust
to bulk PILS mass concentrations of Na$^+$ and Ca$^{2+}$ (see Sect. S1 in the Supplement for additional
information about these equations, Table S1 for variable nomenclature, and Table S2 for values of
constant parameters [e.g., mass ratios]).

$$Na^+_{bulk} = ssNa^+ + Na^+_{dust} \qquad\qquad 1$$

$$Ca^{2+}_{bulk} = ssCa^{2+} + Ca^{2+}_{dust} \qquad\qquad 2$$

$$ssCa^{2+} = ssNa^+ \cdot \left(\frac{Ca^{2+}}{Na^+}\right)_{ss} \qquad\qquad 3$$

$$Ca^{2+}_{dust} = Na^+_{dust} \cdot \left(\frac{Ca^{2+}}{Na^+}\right)_{dust} \qquad\qquad 4$$



$$ssNa^+ = \frac{Ca^{2+}_{bulk} - Na^+_{bulk} \cdot \left(\frac{Ca^{2+}}{Na^+}\right)_{dust}}{\left(\frac{Ca^{2+}}{Na^+}\right)_{ss} - \left(\frac{Ca^{2+}}{Na^+}\right)_{dust}} \tag{5}$$

324 We then use an analogous set of equations (Eqs. 6 – 14) to explore if various combustion
325 processes contribute nonnegligible amounts of $Na^+$ to bulk PILS $Na^+$ mass concentrations (see
326 Sect. S2 for more information).

$$Na^+_{bulk} = ssNa^+ + Na^+_{dust} + Na^+_{comb} \tag{6}$$

$$Ca^{2+}_{bulk} = ssCa^{2+} + Ca^{2+}_{dust} \tag{7}$$

$$K^+_{bulk} = ssK^+ + K^+_{dust} + K^+_{comb} \tag{8}$$

$$ssCa^{2+} = ssNa^+ \cdot \left(\frac{Ca^{2+}}{Na^+}\right)_{ss} \tag{9}$$

$$Ca^{2+}_{dust} = Na^+_{dust} \cdot \left(\frac{Ca^{2+}}{Na^+}\right)_{dust} \tag{10}$$

$$ssK^+ = ssNa^+ \cdot \left(\frac{K^+}{Na^+}\right)_{ss} \tag{11}$$

$$K^+_{dust} = Ca^{2+}_{dust} \cdot \left(\frac{K^+}{Ca^{2+}}\right)_{dust} \tag{12}$$

$$Na^+_{comb} = K^+_{comb} \cdot \left(\frac{Na^+}{K^+}\right)_{comb} \tag{13}$$

$$ssNa^+ = \frac{Na^+_{bulk} - K^+_{bulk} \cdot \left(\frac{Na^+}{K^+}\right)_{comb} + Ca^{2+}_{bulk} \cdot \left[\left(\frac{K^+}{Ca^{2+}}\right)_{dust} \cdot \left(\frac{Na^+}{K^+}\right)_{comb} - \left(\frac{Na^+}{Ca^{2+}}\right)_{dust}\right]}{1 - \left[\left(\frac{Ca^{2+}}{Na^+}\right)_{ss} \cdot \left(\frac{K^+}{Ca^{2+}}\right)_{dust} \cdot \left(\frac{Na^+}{K^+}\right)_{comb}\right] - \left[\left(\frac{K^+}{Na^+}\right)_{ss} \cdot \left(\frac{Na^+}{K^+}\right)_{comb}\right] - \left[\left(\frac{Ca^{2+}}{Na^+}\right)_{ss} \cdot \left(\frac{Na^+}{Ca^{2+}}\right)_{dust}\right]} \tag{14}$$


328 Combustion-generated particles over the NWA can stem from a range of seasonal and
329 perennial processes, each with a different $Na^+$ and $K^+$ emission factor. We use empirical, literature-
330 based values of $\left(\frac{Na^+}{K^+}\right)_{comb}$ for particles emitted from the following combustion-related
331 activities/phenomena: agricultural burning, forest fires, industrial operations, sauna stove wood
332 burning for residential heating, car driving, and coal burning for electricity generation (Table S3).
333 Note that only one value at a time can be used for $\left(\frac{Na^+}{K^+}\right)_{comb}$ in Eqs. 13 and 14, which forces the
334 assumption that all combustion-generated particles collected in PILS samples are from the same
335 source and/or have the same $\left(\frac{Na^+}{K^+}\right)_{comb}$ value.

336 Mass concentrations of $ssNa^+$ determined either by Eqs. 1 – 5 or Eqs. 6 – 14 are then used to
337 determine sea salt mass concentrations (Eq. 15) as well as quantities relevant to $Cl^-$ depletion (Eqs.
338 16 – 26).



$$Sea\ salt = \ ssNa^+ \cdot \left(\frac{total\ mass}{Na^+}\right)_{ss} \qquad 15$$

$$\%Cl^-\ depletion = 100 \cdot \frac{ssNa^+ \cdot \left(\frac{Cl^-}{Na^+}\right)_{ss} - Cl^-_{bulk}}{ssNa^+ \cdot \left(\frac{Cl^-}{Na^+}\right)_{ss}} \qquad 16$$

$$Lost\ Cl^- = ssNa^+ \cdot \left(\frac{Cl^-}{Na^+}\right)_{ss}\ -\ Cl^-_{bulk} \qquad 17$$

$$Lost\ Cl^-_{bulk} = Na^+_{bulk} \cdot \left(\frac{Cl^-}{Na^+}\right)_{ss}\ -\ Cl^-_{bulk} \qquad 18$$

$$Lost\ Cl^-_{diff} = Lost\ Cl^-_{bulk}\ -\ Lost\ Cl^- \qquad 19$$

$$nssSO_4^{2-} = \ SO^{2-}_{4,\,bulk} - ssNa^+ \cdot \left(\frac{SO_4^{2-}}{Na^+}\right)_{ss} \qquad 20$$

$$ExSO_4^{2-} = nssSO_4^{2-} - \frac{MW_{SO_4^{2-}}}{MW_{NH_4^+}} \cdot \frac{NH^+_{4,bulk}}{y_{SO_4^{2-}}} \qquad 21$$

$$ExNH_4^+ = NH^+_{4,bulk} - \frac{MW_{NH_4^+}}{MW_{SO_4^{2-}}} \cdot y_{SO_4^{2-}} \cdot nssSO_4^{2-} \qquad 22$$

$$ExNO_3^- = NO^-_{3,\,bulk} - \frac{MW_{NO_3^-}}{MW_{NH_4^+}} \cdot \frac{ExNH_4^+}{y_{NO_3^-}} \qquad 23$$

$$Excess\ acidic\ species = ExSO_4^{2-} + ExNO_3^- + oxalate_{bulk} \qquad 24$$

$$Lost\ Cl^-\ attr.\ to\ A = \ [A] \cdot y_A \cdot \frac{MW_{Cl^-}}{MW_A} \qquad 25$$

$$Lost\ Cl^-\ attr.\ to\ \text{excess acidic species} = \sum_{A=\ ExSO_4^{2-},\,ExNO_3^-,\,oxalate_{bulk}} Lost\ Cl^-\ attr.\ to\ A \qquad 26$$


We first calculate the percentage of Cl⁻ originally in sea salt particles that has been displaced
by acidic species (%Cl⁻ depletion; Eq. 16) to facilitate comparisons between our results and other
studies. Subsequently, mass concentrations of displaced Cl⁻ are calculated using two approaches
to explore the effects of accounting for non-sea salt sources of Na⁺: Approach 1 quantifies
displaced Cl⁻ using derived mass concentrations of ssNa⁺ (lost Cl⁻; Eq. 17), while Approach 2
determines displaced Cl⁻ using bulk PILS Na⁺ mass concentrations (lost Cl⁻_bulk; Eq. 18), thus
assuming sea salt is the only source of Na⁺. Mass concentrations of lost Cl⁻_bulk will always be
greater than corresponding values of lost Cl⁻, and differences between the two (lost Cl⁻_diff; Eq. 19)
are used to assess the significance in accounting for non-sea salt sources of Na⁺ when evaluating
the extent of Cl⁻ depletion processes and their potential effects on atmospheric chemistry.





As mentioned above, acidic species are responsible for displacing Cl⁻ from sea salt particles.
However, only a subset of the bulk PILS mass concentrations of $SO_4^{2-}$ and $NO_3^-$ are available for
Cl⁻ depletion reactions, as (i) $SO_4^{2-}$ is a naturally occurring component of sea salt and (ii) available
$NH_4^+$ will neutralize certain amounts of $SO_4^{2-}$ and potentially $NO_3^-$, leaving them relatively
unreactive. Equations 20 – 23 determine mass concentrations of non-sea salt, unneutralized $SO_4^{2-}$
, and $NO_3^-$, which are added to bulk PILS mass concentrations of oxalate to quantify the amount
of excess acidic species (Eq. 24) available for displacing Cl⁻ from sea salt particles. Note that we
use oxalate here as a proxy variable to represent organic acids in general as it is typically the most
abundant organic acid in tropospheric aerosol particles (e.g., Hilario et al., 2021; Ziemba et al.,
2011; Cruz et al., 2019). We calculate the theoretical amount of lost Cl⁻ attributable to each excess
acidic species (Eq. 25) as well as the total amount attributed to all measured excess acidic species
(Eq. 26). Results from Eq. 26 can be compared to values from Eq. 17 to identify the amount of lost
Cl⁻ explained by the measured excess acidic species, and discrepancies in these values may indicate
there are additional species contributing to Cl⁻ depletion (e.g., weak organic acids [Laskin et al.,
2012]; reactions initiated by $O_3$ [Keene et al., 1990]).
**2.5 MERRA-2 and NAAPS reanalysis products**
Wind speed and wind direction at 950 hPa were obtained from the Modern-Era Retrospective
Analysis for Research and Application, Version 2 (MERRA-2; Gelaro et al., 2017) to provide
context for large-scale boundary layer wind patterns over the region during each season/category
and/or flights of interest. Monthly averages were attained for December 2021 and January,
February, March, May, and June 2022 at $0.5° \times 0.625°$ spatial resolution, while 3-hour averages
were acquired for periods pertinent to each transit flight as well as the case study flights discussed
in Sects. 3.2 and 3.7.1. Monthly averages for December 2021, January 2022, and February 2022
were combined and averaged to produce a single wind vector field representative of the December-
February category, while averages for March, May, and June 2022 are used to portray conditions
for the March, May, and June Bermuda categories, respectively. The 950 hPa pressure layer was
selected as this is the Falcon's median pressure altitude during the Winter 2022 and Summer 2022
deployments.
We relied on the Navy Aerosol Analysis and Prediction System (NAAPS) to identify the
presence of surface-level dust and smoke over the region on selected days using images from the
Aerosol Modeling archive (https://www.nrlmry.navy.mil/aerosol/) for the "Eastern United States"
and "Tropical Atlantic." We selected images at 1800Z for each day as this time is most relevant to
flights during the Winter 2022 and Summer 2022 deployments. NAAPS surface dust and smoke
mass concentrations are gridded reanalysis products available at $1° \times 1°$ spatial resolution and 6-
hourly temporal resolution, where simulations of dust depend on surface erodible fraction and
surface friction velocity (Lynch et al., 2016), and those of smoke depend on size and duration of
satellite-detected hotspots (Reid et al., 2009; Hyer et al., 2013). Modeled atmospheric transport of
dust and smoke particles is then governed by the Navy Global Environmental Model (NAVGEM;
Hogan et al., 2014). These products are used to explore how influence from dust and smoke plumes
may affect calculations of Cl⁻ depletion for case studies presented in Sect. 3.7.1.
**3.  Results and discussion**
**3.1 Meteorological context**



Meteorological conditions during the Winter 2022 and Summer 2022 deployments are mostly consistent with climatological characteristics reported for the NWA in Sorooshian et al. (2020) and Painemal et al. (2021). Median temperatures are lowest in December-February (2.7° C) followed by March (9.2° C), March transit (13.3° C; recall the March transit flights are in late March), May transit (19.2° C), May (19.9° C), and June Bermuda (21.9° C; Fig. 1). Median water vapor mixing ratios and RH follow the same trend with the exception that RH slightly decreases from December-February (53%) to March (50%) and March transit (47%). Median wind speeds are highest for March (10.3 m s$^{-1}$), similar for December-February, March transit, and May (8.9, 8.6, and 8.4 m s$^{-1}$, respectively), and lowest for May transit and June Bermuda (6.7 and 6.4 m s$^{-1}$, respectively). MERRA-2 wind fields at 950 hPa (e.g., Fig. 2) show westerly flow along the USEC for December-February that transitions to southwesterly flow for March and March transit, which is a typical progression as the Bermuda High begins to strengthen (Davis et al., 1997). For May and May transit, zonal flow returns north of 34° N while relatively weak southwesterly flow persists to the south. Southwesterly winds dominate for June Bermuda, and large-scale flow patterns across the NWA appear conventional for a fully developed summertime Bermuda High.

Median CO volume mixing ratios are highest for categories sampling solely along the USEC (i.e., December-February [133 ppb], March [141 ppb], and May [124 ppb]) compared to June Bermuda (81 ppb), affirming sampled coastal air masses were most influenced by anthropogenic emissions. We refrain from using CO to compare levels of anthropogenic influence between categories focused on the USEC as CO exhibits seasonal dependence over the NWA (Buchholz et al., 2021). Specifically, peak values are typically observed in early spring due to wintertime accumulation caused by reduced destruction by OH, while increased rates of oxidation by OH over summer lead to minimum concentrations in late summer.

Precipitation is considered in this work as (i) wet scavenging processes remove sea salt particles more efficiently than several other particle types (Galloway et al., 1993), and (ii) strong winds associated with precipitation events can enhance sea salt emissions and offset scavenging losses (Dadashazar et al., 2021; Grandey et al., 2011), both of which can influence the amount of Cl$^-$ available for depletion reactions on shorter time scales than the seasonal factors discussed above. The NWA receives the most rainfall from December – February followed by June – August, with precipitation rates peaking along the Gulf Stream in all seasons (Painemal et al., 2021). Hawcroft et al. (2012) showed that 65 – 80% and 50 – 70% of the rainfall over the NWA in December – February and June – August, respectively, is associated with midlatitude cyclones (MLC)s, a common year-round weather phenomenon for the region (e.g., Braun et al., 2021; Eichler and Higgins, 2006) largely dictating the eastward transport of trace gases and particulates from North America to the adjacent marine environment (Keim et al., 2005; Cooper et al., 2002, 2001). Despite their frequency and known effects on other aerosol properties (e.g., aerosol optical depth and size distribution; Grandey et al., 2011), there is uncertainty in how frontal passages influence parameters relevant to Cl$^-$ depletion over the NWA. During this study, meteorological conditions were often driven by MLCs, with synoptic conditions changing every few days (Table 2). We discuss key variables in the context of prefrontal and postfrontal airstreams associated with MLCs to explore the influence of midlatitude weather disturbances on depletion reactions and Cl radical budgets over the NWA. Finally, note that clear-ensemble data for December-February do not extend eastward of ~73° W due to frequent cloud cover below 3 km over the ocean. This should be taken into consideration when comparing results for December-February to other categories, especially for continentally sourced properties and/or those that depend on wind fetch.



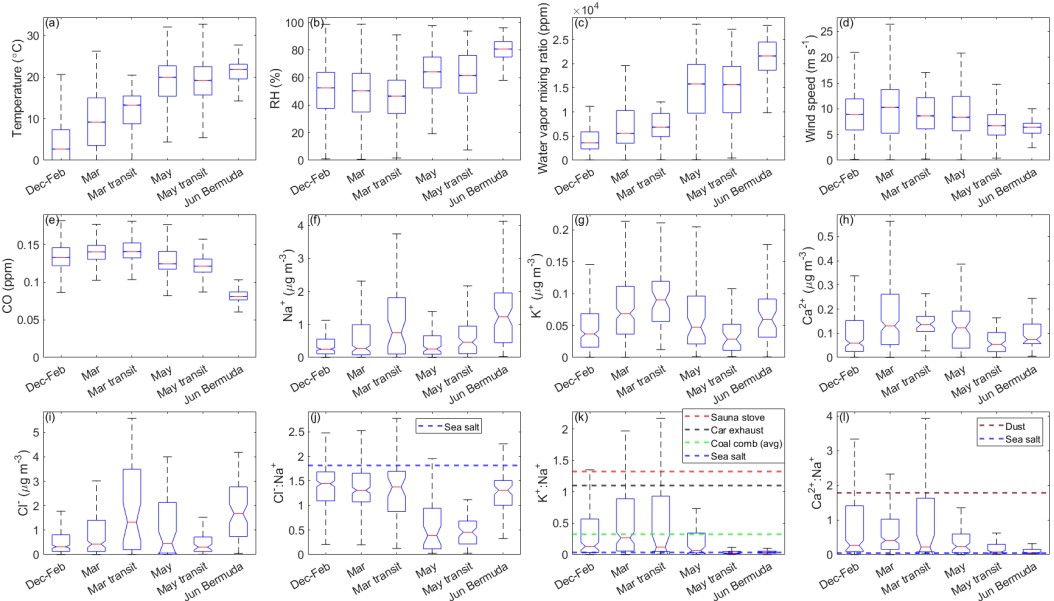

438

**Figure 1.** Notched box plots showing seasonal/categorical differences in (**a**) temperature, (**b**) relative humidity (RH), (**c**) water vapor mixing ratio, (**d**) wind speed, (**e**) carbon monoxide (CO) mixing ratios, bulk mass concentrations from a particle into liquid sampler (PILS) of (**f**) chloride ($Cl^-$), (**g**) sodium ($Na^+$), (**h**) potassium ($K^+$), and (**i**) calcium ($Ca^{2+}$), as well as ratios of these mass concentrations for (**j**) $Cl^-:Na^+$, (**k**) $K^+:Na^+$, and (**l**) $Ca^{2+}:Na^+$. Data are from clear ensembles only. Typical ratios for particular ions in sea salt and/or dust are marked with dashed lines in **j**, **k**, and **l**. In **k**, we use additional lines to indicate ratios of $K^+:Na^+$ reported in the literature for inefficient batch combustion in a sauna stove (1.33; Lamberg et al., 2011), car exhaust (1.1; Huang et al., 1994), and coal combustion (0.33; Ondov et al., 1989). The solid red line in the center of each box indicates the median, box edges represent the 25th and 75th percentiles, and the lower and upper whiskers indicate the lower limit (first quartile - 1.5 × interquartile range) and upper limit (third quartile + 1.5 × interquartile range), respectively. The notches span the 95th confidence interval for the median.

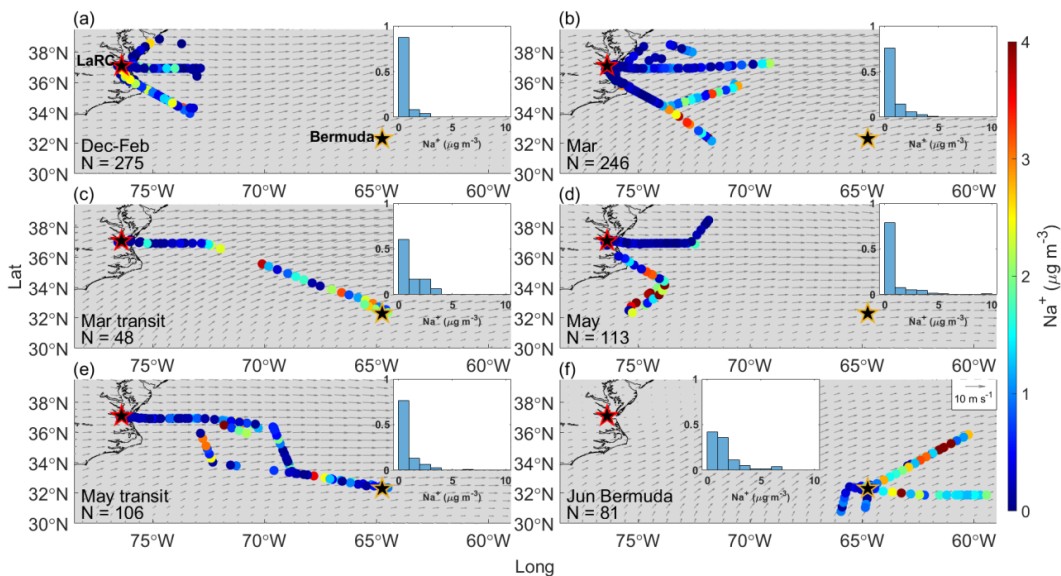

452

**Figure 2.** Bulk PILS $Na^+$ mass concentrations from clear ensembles during (**a**) December 2021-
February 2022, (**b**) March 2022, (**c**) March 2022 transit flights between NASA Langley Research
Center (LaRC; marked with a red-edged star) and Bermuda (marked with a golden-edged star), (**d**)
May 2022, (**e**) May 2022 transit flights between LaRC and Bermuda, and (**f**) the Bermuda field
campaign in June 2022. Normalized histograms in each panel show the distribution of bulk PILS
$Na^+$ mass concentrations for that specific category since overlap among the colored dots can hide
some from view. Grey arrows indicate the average magnitude and direction of winds at 950 hPa
from MERRA-2 for the period relevant to each category.





**Table 2.** Dates, sample quantities, meteorological conditions, and aerosol particle properties relevant to Cl⁻ depletion for research flights (RFs) considered in each category. Median values of $Na^+_{bulk}$ and $Ca^{2+}_{bulk}$ are based on bulk PILS data while values of lost Cl⁻, Cl⁻ depletion, and excess acidic species are derived using Eqs. 1 – 5, 16, 17, and 20 - 24. "N PILS samples" refers to the total number of PILS samples collected during clear ensembles on the date indicated, while "$N_{PILS}$" refers to the number of these samples providing enough information to determine a given property. "$N_{PILS\&AMS}$" refers to the number of coinciding mass concentrations from the PILS and aerosol mass spectrometer (AMS) necessary for calculating excess acidic species mass concentrations.

| Category | Date | RF(s) | N PILS samples | Meteorological conditions and/or relevant notes | $Na^+_{bulk}$ | | $Ca^{2+}_{bulk}$ | | Lost Cl⁻ | | Cl⁻ depletion | | Excess acidic species | |
|---|---|---|---|---|---|---|---|---|---|---|---|---|---|---|
| | | | | | Median ($\mu g\ m^{-3}$) | $N_{PILS}$ | Median ($\mu g\ m^{-3}$) | $N_{PILS}$ | Median ($\mu g\ m^{-3}$/ pptv) | $N_{PILS}$ | Median (%) | $N_{PILS}$ | Median ($\mu g\ m^{-3}$) | $N_{PILS\ \&\ AMS}$ |
| Dec-Feb | 30 November 2021 | 94 | 7 | Remains of post-frontal conditions | 0.14 | 7 | 0.31 | 7 | -0.17/NA[1] | 7 | 0 | 7 | 0.29 | 13 |
| | 01 December 2021 | 95 | 16 | Prefrontal, high pressure; smoke in boundary layer near coast | 0.30 | 16 | 0.49 | 16 | -0.16/NA[1] | 16 | 0 | 16 | 0.59 | 136 |
| | 07 December 2021 | 96 | 5 | Postfrontal, cold high pressure behind a strong cold front | 0.19 | 5 | 0.20 | 5 | -0.12/NA[1] | 5 | 0 | 5 | 0.03 | 22 |
| | 11 January 2022 | 100, 101 | 6 | Cold high pressure, cold air outbreak (CAO) conditions | 0.34 | 4 | 0.05 | 6 | 0.12/80 | 4 | 20 | 4 | 0.49 | 21 |
| | 12 January 2022 | 102, 103 | 33 | Cold high pressure | 0.21 | 29 | 0.06 | 21 | 0.01/7 | 15 | 4 | 15 | 0.20 | 109 |
| | 15 January 2022 | 104 | 3 | Postfrontal | 0.63 | 3 | 0.05 | 2 | 0.01/7 | 2 | 4 | 2 | 0.35 | 20 |
| | 18 January 2022 | 105 | 11 | Low pressure moves offshore, sets up CAO conditions | 0.22 | 2 | 0.06 | 2 | NaN | 0 | NaN | 0 | 0.01 | 10 |
| | 19 January 2022 | 107, 108 | 26 | Short-lived high pressure | 0.24 | 14 | 0.06 | 10 | -0.05/NA[1] | 6 | 0 | 6 | 0.14 | 66 |
| | 24 January 2022 | 109, 110 | 26 | Postfrontal, weak high pressure | 0.07 | 15 | 0.03 | 13 | -0.04/NA[1] | 8 | 0 | 8 | 0.02 | 86 |
| | 26 January 2022 | 111, 112 | 20 | Postfrontal | 0.12 | 12 | 0.03 | 10 | 0.00/0 | 7 | 0 | 7 | 0.01 | 83 |
| | 27 January 2022 | 113, 114 | 18 | Cold high pressure | 0.25 | 16 | 0.01 | 5 | 0.06/40 | 5 | 21 | 5 | 0.36 | 41 |
| | 01 February 2022 | 115 | 8 | High pressure | 0.90 | 6 | 0.05 | 7 | 0.41/273 | 5 | 21 | 5 | 1.00 | 37 |
| | 02 February 2022 | 116 | 17 | High pressure | 0.73 | 16 | 0.03 | 6 | 0.18/120 | 6 | 12 | 6 | 0.41 | 44 |
| | 03 February 2022 | 117, 118 | 15 | High pressure | 1.03 | 14 | 0.03 | 5 | 0.04/27 | 5 | 2 | 5 | 0.00 | 10 |
| | 15 February 2022 | 120, 121 | 34 | Postfrontal conditions, cold high pressure | 0.25 | 27 | 0.03 | 24 | 0.08/53 | 21 | 17 | 21 | 0.56 | 69 |
| | 16 February 2022 | 122, 123 | 21 | Cold high pressure | 0.20 | 18 | 0.08 | 20 | 0.10/67 | 16 | 27 | 16 | 0.53 | 105 |
| | 19 February 2022 | 124, 125 | 38 | Weak postfrontal | 0.12 | 30 | 0.06 | 37 | 0.06/40 | 23 | 24 | 23 | 0.24 | 186 |



| Month | Date | Flight | Count | Description | | | | | | | | | | |
|---|---|---|---|---|---|---|---|---|---|---|---|---|---|---|
| | 22 February 2022 | 126, 127 | 25 | Prefrontal, high pressure | 1.41 | 25 | 0.12 | 24 | 0.45/300 | 24 | 17 | 24 | 0.64 | 184 |
| | 26 February 2022 | 128, 129 | 16 | Postfrontal | 0.13 | 16 | 0.06 | 15 | -0.02/NA[1] | 15 | 0 | 15 | 0.27 | 130 |
| | Overall | | 345 | | 0.25 | 275 | 0.06 | 235 | 0.04/27 | 190 | 6 | 190 | 0.30 | 1372 |
| Mar | 02 March 2022 | 130 | 39 | Postfrontal, high pressure | 0.30 | 36.00 | 0.16 | 39 | 0.04/27 | 33 | 8 | 33 | 1.20 | 298 |
| | 03 March 2022 | 131, 132 | 71 | Weak prefrontal | 0.91 | 57.00 | 0.27 | 71 | 0.10/67 | 57 | 9 | 57 | 1.19 | 537 |
| | 04 March 2022 | 133, 134 | 42 | Cold high pressure | 1.56 | 40.00 | 0.12 | 39 | 0.42/280 | 36 | 14 | 36 | 1.02 | 242 |
| | 13 March 2022 | 138 | 8 | Postfrontal, CAO conditions | 0.12 | 6.00 | 0.06 | 7 | -0.12/NA[1] | 6 | 0 | 6 | 0.02 | 22 |
| | 14 March 2022 | 139, 140 | 38 | Late postfrontal, cold high pressure; smoke plume sampled from a woodland fire | 0.16 | 37.00 | 0.06 | 37 | 0.03/20 | 35 | 13 | 35 | 0.22 | 305 |
| | 18 March 2022 | 141 | 14 | Weak postfrontal | 0.18 | 14.00 | 0.04 | 12 | 0.05/33 | 12 | 35 | 12 | 0.33 | 98 |
| | 26 March 2022 | 144, 145 | 29 | Postfrontal; sampled dust, smoke, and potentially pollen | 0.05 | 22.00 | 0.04 | 22 | -0.02/NA[1] | 13 | 0 | 13 | 0.00 | 147 |
| | 28 March 2022 | 146 | 17 | Postfrontal | 0.07 | 17.00 | 0.05 | 12 | -0.01/NA[1] | 10 | 0 | 10 | 0.13 | 98 |
| | 29 March 2022 | 147, 148 | 19 | Postfrontal, high pressure, CAO conditions | 0.21 | 17.00 | 0.05 | 5 | 0.02/13 | 4 | 34 | 4 | 0.00 | 43 |
| | Overall | | 277 | | 0.27 | 246 | 0.13 | 244 | 0.04/27 | 206 | 10 | 206 | 0.57 | 1790 |
| May | 03 May 2022 | 149 | 15 | Weak prefrontal; presence of smoke potentially from New Mexico | 0.42 | 15 | 0.14 | 12 | 0.89/594 | 7 | 85 | 7 | 0.03 | 92 |
| | 05 May 2022 | 150, 151 | 18 | Postfrontal | 0.05 | 14 | 0.04 | 14 | 0.42/280 | 2 | 89 | 2 | 0.02 | 91 |
| | 16 May 2022 | 153, 154 | 39 | Prefrontal to an approaching cold front yet also postfrontal to a departing band of precipitation | 0.26 | 39 | 0.26 | 7 | 0.65/434 | 1 | 73 | 1 | 0.05 | 85 |
| | 17 May 2022 | 155 | 37 | Postfrontal | 0.08 | 17 | 0.01 | 13 | 1.53/1020 | 2 | 73 | 2 | 0.05 | 52 |
| | 20 May 2022 | 158 | 28 | Warm high pressure, southerly flow due to Bermuda high[2]; haze with potential sampling of bioaerosol | 1.75 | 28 | 0.17 | 27 | 1.91/1274 | 21 | 48 | 21 | 0.97 | 148 |
| | Overall | | 137 | | 0.26 | 113 | 0.12 | 73 | 1.76/1174 | 33 | 64 | 33 | 0.05 | 468 |
| Mar transit | 22 March 2022 | 142, 143 | 48 | High pressure, two days after a cold front and two days before another cold front | 0.75 | 48 | 0.14 | 48 | 0.11/73 | 43 | 9 | 43 | 0.36 | 423 |
| May transit | 18 May 2022 | 156, 157 | 67 | Postfrontal along East Coast, aircraft passed across the cold front on the way to Bermuda | 0.51 | 58 | 0.05 | 50 | 1.37/914 | 31 | 74 | 31 | 0.27 | 216 |




| | Date | | | Description | | | | | | | | | | |
|---|---|---|---|---|---|---|---|---|---|---|---|---|---|---|
| | 21 May 2022 | 159, 160 | 42 | Warm high pressure, anticyclonic flow around Bermuda high | 0.50 | 37 | 0.08 | 26 | 1.67/1114 | 17 | 75 | 17 | 1.87 | 137 |
| | 31 May 2022 | 161 | 11 | Postfrontal | 0.18 | 11 | 0.02 | 5 | 0.22/147 | 5 | 67 | 5 | 0.02 | 20 |
| | **Overall** | | **120** | | **0.46** | **106** | **0.05** | **81** | **1.33/887** | **53** | **74** | **53** | **0.44** | **373** |
| Jun Bermuda | 02 June 2022 | 162, 163 | 4 | Prefrontal | 0.64 | 4 | 0.03 | 3 | 0.71/474 | 2 | 44 | 2 | 2.62 | 12 |
| | 03 June 2022 | 164 | 1 | Prefrontal, tropical system approaching from the southwest | 0.30 | 1 | NaN | 0 | NaN | 0 | NaN | 0 | 0.02 | 1 |
| | 05 June 2022 | 165 | 29 | Could only fly in the morning due to approaching tropical cyclone (TC), TC departs 06 June 2022. | 1.76 | 29 | 0.08 | 26 | 1.35/900 | 26 | 36 | 26 | 1.97 | 213 |
| | 07 June 2022 | 167 | 1 | High behind departing TC | 2.21 | 1 | NaN | 0 | NaN | 0 | NaN | 0 | 0.02 | 1 |
| | 08 June 2022 | 168, 169 | 2 | High pressure behind TC, African dust known to be in domain | 4.28 | 2 | 1.07 | 1 | 1.12/747 | 1 | 11 | 1 | 0.04 | 9 |
| | 10 June 2022 | 170 | 1 | High pressure, isolated thunderstorms, African dust known to be in domain | 2.28 | 1 | 0.06 | 1 | 0.68/454 | 1 | 17 | 1 | 1.19 | 9 |
| | 11 June 2022 | 172, 173 | 20 | High pressure, African dust known to be in domain | 0.33 | 20 | 0.21 | 12 | 0.15/100 | 11 | 11 | 11 | 1.12 | 71 |
| | 13 June 2022 | 174 | 25 | High pressure, African dust known to be in domain but sampled away from dust for contrast | 1.34 | 23 | 0.06 | 24 | 0.48/320 | 23 | 17 | 23 | 1.89 | 170 |
| | **Overall** | | **83** | | **1.24** | **81** | **0.07** | **67** | **0.66/440** | **64** | **25** | **64** | **1.82** | **486** |

[1]Negative mass concentrations in $\mu$g m$^{-3}$ are reported for lost Cl$^-$ and can be conceptualized as the amount of measured particulate Cl$^-$ in excess of what would be in unreacted sea salt particles based on Eqs. 1 – 4. Negative values may suggest there are additional non-sea salt sources of particulate Cl$^-$ within the sampled air mass. In these cases, we do not provide corresponding gas phase concentrations of lost Cl$^-$ in pptv as these are only meaningful when Cl$^-$ is displaced from sea salt particles.

[2]Davis et al. (1997)



**3.2 Seasonal, spatial, and frontal trends in Na$^+$**

Cl$^-$ depletion studies are motivated by the fact that radicals produced via depletion reactions can influence atmospheric chemistry, the extent to which largely depends on the quantity of radicals generated. Therefore, the amount of Cl$^-$ in sea salt available to depletion reactions is critical to quantify, which is why a large portion of our initial discussion is about trends in bulk Na$^+$ mass concentrations as they are a reliable indicator of sea salt mass concentrations. Bulk PILS Na$^+$ mass concentrations are remarkably similar for December-February, March, and May (median mass concentrations of 0.25, 0.27, and 0.26 µg m$^{-3}$, respectively), higher for March transit and May transit (0.75 and 0.46 µg m$^{-3}$, respectively), and highest in and around Bermuda (1.24 µg m$^{-3}$). In general, past works also typically report higher sea salt mass concentrations in open-ocean environments compared to coastal locations (Table S4), which is intuitive considering that wind fetch is one important factor governing atmospheric sea salt mass concentrations. However, if Na$^+$ mass concentrations were dictated chiefly by wind fetch over the NWA, values would mostly increase moving eastward, which is not always the case (e.g., Fig. 2e). In fact, there does not appear to be any distinct spatial gradients in Na$^+$ mass concentrations for the seasons/categories presented, yet (i) overlap of flight tracks makes it difficult to view all mass concentrations at once, and (ii) we do not have enough data to state that this is always true for the region.

Aside from wind fetch, removal via wet scavenging processes is another factor dictating sea salt mass concentrations over marine environments. We explore the effect of passing frontal systems on bulk Na$^+$ mass concentrations for December-February, March, and May as (i) bulk Na$^+$ appears seasonally independent among these categories and (ii) flights sampled the same general region, allowing us to remove coastal versus open-ocean sampling as a confounding variable. When applying the meteorological conditions identified for each day in Table 2, bulk Na$^+$ mass concentrations are generally higher during prefrontal/high pressure conditions compared to postfrontal scenes for each seasonal/monthly category (Fig. 3). It is not unusual for bulk Na$^+$ mass concentrations to exceed 3 µg m$^{-3}$ in prefrontal and/or high-pressure conditions, especially in March and May, yet values never exceed this threshold in postfrontal conditions. Although bulk statistics suggest frontal passages may reduce sea salt mass concentrations over the NWA, data from prefrontal and postfrontal conditions are not guaranteed to be linked, meaning samples quantifying bulk Na$^+$ before and after each frontal passage are not always available. Therefore, we isolate bulk Na$^+$ mass concentrations for flights straddling frontal passages to assess the relationship of sea salt mass concentrations and MLCs on a case-study level.

Postfrontal conditions on 19 February 2022 (RFs 124 and 125) are associated with bulk Na$^+$ mass concentrations mostly < 0.3 µg m$^{-3}$ and moderate westerly winds bringing continental air over the NWA (Fig. S1). Three days later (22 February 2022; RFs 126 and 127), prefrontal conditions show increased bulk Na$^+$ mass concentrations that are distributed evenly from 0.3 – 2.7 µg m$^{-3}$ and southerly winds along the coast. Bulk Na$^+$ mass concentrations then swiftly decrease to values mostly below 0.3 µg m$^{-3}$ by 26 February 2022 (RFs 128 and 129) as another MLC moves through the region, although note clear-ensemble sampling was more restricted to the coastline on this day compared to 19 and 22 February 2022.

Unfortunately, samples straddling a frontal passage for March are unavailable, but we use consecutive flights from 02 – 04 March 2022 to depict the "recharge" of sea salt mass concentrations following a MLC (Fig. S2). Bulk Na$^+$ mass distributions gradually shift towards larger mass concentrations moving from postfrontal conditions with gentle westerly winds (02 March 2022; RF 130), to weak prefrontal conditions with stronger northwesterly and southwesterly winds converging at 36 °N (03 March 2022; RFs 131 and 132), and, finally, to cold high-pressure





conditions with moderate anticyclonic flow around a high over the northeastern U.S. (04 March 2022; RFs 133 and 134). Air masses sampled on 02 and 03 March 2022 appear more continentally influenced and may have been more recently affected by large-scale precipitation compared to the marine air mass sampled on 04 March 2022, which is a potential explanation for the differences in bulk $Na^+$ mass concentrations.

Flights on 16 May 2022 (RFs 153 and 154) sampled an air mass recently impacted by a retreating band of precipitation yet also considered as prefrontal due to an approaching cold front (Fig. S3). As there was limited time for sea salt mass concentrations to recharge between the consecutive MLCs, it is unsurprising there is little difference in bulk $Na^+$ mass concentrations between 16 May 2022 and the postfrontal conditions sampled on 17 May 2022 (RF 155). Frontal influence dissipated by 20 May 2022 (RF 158) with southwesterly flow returning along the coastline in association with the strengthening Bermuda High. This, and the absence of precipitation for several days, may help explain the increase in bulk $Na^+$ mass concentrations from mostly below 1 µg m$^{-3}$ on 16 – 17 May 2022 to a mostly above this value on 20 May 2022. The three case studies presented are meant to illustrate how rapidly sea salt mass concentrations can change over the NWA due, in part, to fluctuations in synoptic-scale wind patterns and/or large-scale precipitation associated with MLCs. However, we acknowledge that there are many other confounding atmospheric variables influencing sea salt mass concentrations during these case studies and that flight tracks do not cover the exact same locations on each of these days. Although we do not have enough data to make definitive claims, bulk statistical and case study analyses suggest sea salt mass concentrations decrease behind passing MLCs over the NWA, which corresponds to reduced potential in the amount of reactive chlorine-containing gases that could be produced via depletion reactions compared to in prefrontal and high-pressure conditions.

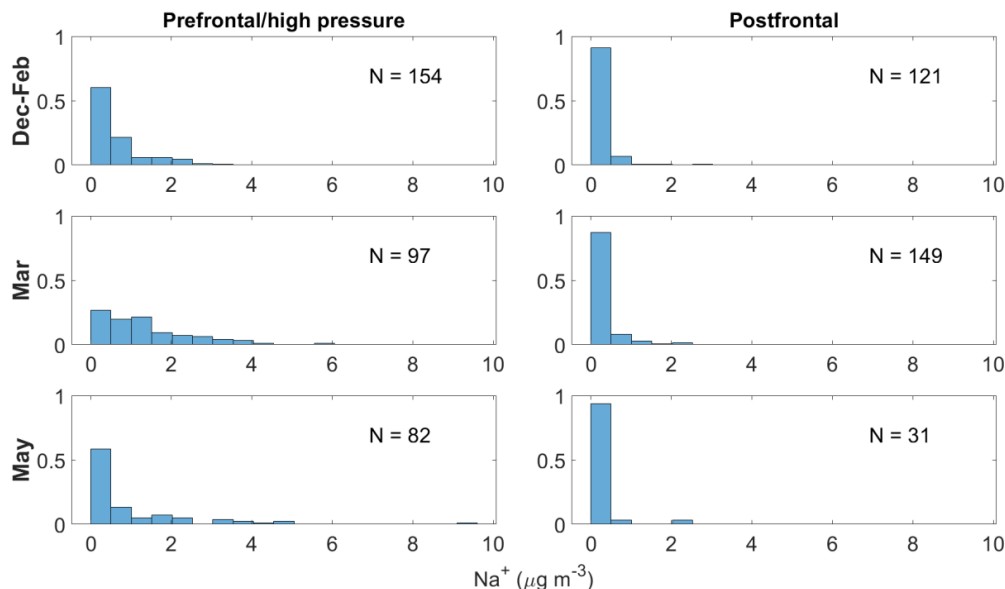

**Figure 3.** Normalized histograms showing differences in bulk PILS $Na^+$ mass concentrations from clear ensembles occurring in prefrontal and/or high-pressure versus postfrontal conditions for December-February (top row), March (middle row), and May (bottom row). These categories are shown as they represent flights occurring in and around the East Coast, eliminating coastal versus open-ocean sampling as a confounding variable.



**3.3 Seasonal trends in K⁺, Ca²⁺, Cl⁻ and ion mass ratios**

As described above, the NWA receives BB emissions from continuous sources (e.g., fossil fuel combustion for transportation and industrial efforts along the USEC), seasonal practices (e.g., agricultural waste burning in spring, wood burning in winter), and intermittent yet influential events (e.g., forest fires). Using $K^+$ as a tracer for such activities, BB influence is greatest during March and March transit flights with median bulk $K^+$ mass concentrations of 0.07 and 0.09 µg m⁻³, respectively, compared to 0.04, 0.05, 0.03 and 0.06 µg m⁻³ for the December-February, May, May transit, and June Bermuda categories, respectively. This agrees with previous findings where mass concentrations of organic carbon and particles with diameters $2.5 - 10$ µm ($PM_{coarse}$) were much higher in March than in any other month at a coastal site in Florida (Edwards et al., 2021), and this was attributed mostly to the annual peak in prescribed burning across the southeastern U.S. (Jaffe et al., 2020; McCarty et al., 2007). Our bulk $K^+$ mass concentrations are comparable to mean values reported at a receptor site for BB and urban emissions from East Asia ($0.02 - 0.05$ µg m⁻³; Boreddy and Kawamura, 2015) as well as those in polluted air masses containing dust (0.03 µg m⁻³) and biogenically influenced air masses (0.03 µg m⁻³) over the southeastern U.S. during the Study of Emissions and Atmospheric Composition, Clouds, and Climate Coupling by Regional Surveys (SEAC⁴RS; Kacenelenbogen et al., 2022). However, bulk $K^+$ values are mostly lower than average $K^+$ mass concentrations in air masses influenced by agricultural burning (0.10 µg m⁻³) and wildfire emissions (0.09 µg m⁻³) during SEAC⁴RS (Kacenelenbogen et al., 2022) and also lower than average mass concentrations (0.82 µg m⁻³) measured during the Fire Influence on Regional to Global Environments and Air Quality (FIREX-AQ) airborne field campaign (Adachi et al., 2022) sampling BB plumes in the western and southeastern U.S. Thus, BB particles were consistently present during the Winter 2022 and Summer 2022 deployments, yet relatively dilute compared to their levels in air masses more heavily influenced by BB processes. This is an important point to consider when contemplating how BB emissions may affect estimates of $Cl^-$ depletion, which is discussed in greater detail in Sect. 3.7.2.

We use bulk $Ca^{2+}$ to identify influence from dust particles and see a similar trend as above where median bulk $Ca^{2+}$ mass concentrations are higher in certain spring categories (0.13, 0.14, and 0.12 µg m⁻³ for the March, March transit, and May categories, respectively) compared to December-February (0.06 µg m⁻³) and June Bermuda (0.07 µg m⁻³). Higher springtime bulk $Ca^{2+}$ mass concentrations are likely due to periodic influence from Asian dust plumes, which arrive most frequently over the region from March-May (Aldhaif et al., 2020), and/or to increased suspension of dust particles in BB plumes from agricultural fires across the eastern and southeastern U.S. due to turbulent mixing around flames and the burn front (e.g., Kavouras et al., 2012; Popovicheva et al., 2014; Maudlin et al., 2015; Schlosser et al., 2017; Palmer, 1981). Interestingly, bulk $Ca^{2+}$ mass concentrations are lowest for May transit (0.05 µg m⁻³), but this may be explained by the episodic nature of dust events over the NWA (e.g., Wu et al., 2015; Perry et al., 1997; Prospero, 1999) and the fact that this category is comprised of only three days. African dust plumes become more common over the NWA from June-August (Zuidema et al., 2019) with the strengthening of the Bermuda High, yet the Summer 2022 deployment ended just as these plumes were becoming evident over the region (see meteorological notes for 10, 11, and 13 June 2022 in Table 2). There does not appear to be distinct spatial trends in bulk $Ca^{2+}$ over the region for most categories (Fig. S4), presumably as fluctuations in bulk $Ca^{2+}$ may be largely driven by periodic influence from long-range dust transport, smoke plumes from fires along the USEC advecting over the ocean, and midlatitude weather disturbances (Fig. S5). However, a gradient seems to exist along the March transit flights (RFs 142 and 143 on 22 March 2022) such that bulk





$Ca^{2+}$ mass concentrations are highest to the east of LaRC and then decrease to the southeast
towards Bermuda. This potential sampling of a dust plume and its implications on calculations
relevant to $Cl^-$ depletion are explored further in Sect. 3.7.1.
Median $Cl^-$ mass concentrations exhibit slightly different seasonal trends than bulk $Na^+$, with
values lowest for May transit (0.31 µg m$^{-3}$), slightly higher for December-February, March, and
May (0.32, 0.43, and 0.46 µg m$^{-3}$, respectively), and much higher for March transit and Bermuda
(1.33 and 1.68 µg m$^{-3}$, respectively). The fact that May transit has the third highest median bulk
$Na^+$ mass concentration yet the lowest $Cl^-$ median is the main difference in seasonal trends between
these species, which may seem to suggest $Cl^-$ depletion processes are most active for May transit.
However, the number of PILS samples providing (i) bulk $Na^+$ and (ii) $Cl^-$ mass concentrations are
very different for May (113 and 43, respectively) and May transit (106 and 65, respectively), yet
comparable for December-February, March, March transit, and June Bermuda (Table S5). Thus,
it is best to avoid drawing conclusions about $Cl^-$ depletion from individual trends in bulk $Na^+$ and
$Cl^-$, and to instead focus on samples providing mass concentrations for both species. These samples
were isolated to generate the statistics shown in Fig. 1j, which (i) can be considered as a precursory
analysis for $Cl^-$ depletion over the NWA where sea salt is assumed to be the only source of $Na^+$,
and (ii) are directly comparable to many past works making this assumption. Ratios of $Cl^-$:$Na^+$ are
below 1.81 for all categories, suggesting $Cl^-$ depletion processes are consistently occurring over
the region. However, median values are much lower for May (0.39) and May transit (0.46)
compared to December-February (1.44), March (1.31), March transit (1.38), and June Bermuda
(1.31), suggesting that depletion reactions are particularly prevalent in late spring. May and May
transit ratios are comparable to those previously reported along the USEC (Quinn and Bates, 2005;
Nolte et al., 2008; Zhao and Gao, 2008) in late spring and summer, especially for submicron sea
salt particles.
As mentioned above, $Cl^-$:$Na^+$ ratios are only an appropriate means to illustrate the extent of $Cl^-$
depletion if sea salt is the predominant source of each species. Ratios of bulk $K^+$:$Na^+$ and $Ca^{2+}$:$Na^+$
are useful for indicating if other particle types may be contributing to bulk $Na^+$ concentrations as
these ions are present in distinctly different proportions in sea salt, emissions from various
combustion processes, and dust particles. Combustion and/or BB activities do not appear to
contribute meaningfully to bulk $Na^+$ for May, May transit, and June Bermuda as $K^+$:$Na^+$ ratios
(0.065, 0.020, and 0.037, respectively) are fairly similar to the reference value for sea salt (0.036;
Seinfeld and Pandis, 2016; Finlayson-Pitts and Pitts, 2000), whereas ratios exceeding this value
are observed for December-February (0.132), March (0.267), and March transit (0.119). Table 2
indicates smoke was only directly sampled on four days of the Winter 2022 and Summer 2022
deployments (01 December 2021, 14 March 2022, 26 March 2022, and 03 May 2022), suggesting
increased $K^+$:$Na^+$ ratios for December-February, March, and March transit may have been driven
by increased background levels of BB particles over the NWA from widespread and continuous
residential wood burning and prescribed agricultural burning in winter and early spring as opposed
to acute BB events. All categories have median $Ca^{2+}$:$Na^+$ ratios exceeding the reference value for
sea salt (0.038; Bowen, 1979; Finlayson-Pitts and Pitts, 2000), with values of 0.412, 0.261, 0.233,
0.219, 0.075, and 0.050 for March, December-February, May, March transit, May transit, and June
Bermuda, respectively. These results nicely motivate an investigation into how estimates of Cl-
depletion change when eliminating contributions of (i) dust and (ii) both dust and combustion
emissions to bulk $Na^+$ mass concentrations, which are the topics of Sects. 3.7.1 and 3.7.2,
respectively.





### 3.4 Seasonal, spatial, and frontal trends in acidic species

Sea salt mass concentrations alone control the maximum amount of reactive chlorine-containing gases that can be released via $Cl^-$ depletion reactions, but available acidic species are an important factor in regulating the extent to which these reactive gases are actually released. Median mass concentrations of bulk $SO_4^{2-}$ show that this acidic species is a common constituent of sampled air masses, especially for March transit and June Bermuda (Fig. 4; Table S6). Median bulk $NO_3^-$ mass concentrations are of similar magnitude to bulk $SO_4^{2-}$, yet exhibit less variability among the categories, while oxalate is present in relatively low amounts for December-February, March, and March transit, increases sharply for May and May transit, and then decreases slightly for June Bermuda. In Sect. 2.4, we describe how $ssNa^+$ mass concentrations and subsequently derived parameters can be calculated either by assuming (i) dust and sea salt or (ii) dust, sea salt, and combustion-sourced particles contribute to bulk $Na^+$. In this section and Sects. 3.5, 3.6, 3.7.1, and 3.7.3, we discuss values based on the first assumption, whereas those based on the second assumption are the topic of Sect. 3.7.2.

After accounting for contributions of sea salt to $SO_4^{2-}$ and neutralization of non-sea salt $SO_4^{2-}$ and $NO_3^-$ with $NH_4^+$, excess $SO_4^{2-}$ ($ExSO_4^{2-}$) is typically nonexistent for all categories except June Bermuda (median of 0.63 µg m$^{-3}$; Fig. S6), while a range of mass concentrations of excess $NO_3^-$ ($ExNO_3^-$) remain for all categories except May (0.24, 0.51, 0.32, 0.74, 1.02 µg m$^{-3}$ for December-February, March, March transit, May transit, and June Bermuda, respectively). Thus, mass concentrations of measured acidic species available to participate in $Cl^-$ depletion reactions are relatively low for May (0.05 µg m$^{-3}$; contributed mostly by oxalate), moderate for December-February, March, March transit, and May transit (0.30, 0.57, 0.36, 0.44 µg m$^{-3}$, respectively), and relatively high for June Bermuda (1.82 µg m$^{-3}$). However, recall that oxalate is used in this study as a proxy for general trends in organic acids, many of which have been shown to considerably displace $Cl^-$ from sea salt particles (e.g., Laskin et al., 2012), including formate, acetate, MSA, and succinate (Kerminen et al., 1998; Braun et al., 2017); thus the results based on oxalate are a lower bound for the effects organic acids have on depletion reactions. Although lower than other aerosol constituents, oxalate mass concentrations are highest for May and May transit along with those of $m/z$ 44, a marker of oxygenated organics that has been shown to correlate with organic acids (Zhang et al., 2005; Takegawa et al., 2007; Sorooshian et al., 2010), and $m/z$ 79, a marker for MSA (Zorn et al., 2008; Van Rooy et al., 2021). Median $m/z$ 44 mass concentrations especially suggest organic acids may play an important role in sea salt particle chemistry for May and May transit as values (0.46 and 0.41 µg m$^{-3}$, respectively) (i) are comparable to those of other dominant acidic species over the region, (ii) represent the mass only of the particle fragments (i.e., carboxylic acids) able to displace $Cl^-$, and (iii) reflect a lower limit of what is actually available for depletion reactions as AMS measurements are for particles 60 – 600 nm.

Like sea salt mass concentrations, excess acidic species do not display clear zonal or meridional trends over the NWA (Fig. S7) but do appear to decrease near the USEC following the passage of MLCs (Fig. S8). The reasons are uncertain for such high mass concentrations of excess acidic species for June Bermuda, but a probable cause may be emissions of DMS from marine organisms oxidizing to produce $H_2SO_4$ (e.g., Luria et al., 1989; Andreae et al., 2003). Excess acidic species mass concentrations are not nearly as high near Bermuda for March transit and May transit compared to June Bermuda, suggesting the increased values in June may be (i) due to greater photochemical production of $SO_4^{2-}$ with increased incident solar radiation (Parungo et al., 1987; Corral et al., 2021) or (ii) due to an episodic surge in local marine biological activity, which has been shown to occur around Bermuda when higher doses of solar radiation become available to



the upper mixed layer of the ocean (Vallina and Simó, 2007; Toole and Siegel, 2004). Level-3 (8-
day average, 4 km resolution) sea surface chlorophyll a concentrations from MODIS-Aqua show
consistent values around Bermuda for March transit, May transit, and June Bermuda. However,
there is an important distinction between biomass and ocean biological activity such that steady
biomass around Bermuda does not necessarily correspond to similar gaseous emission rates for
these categories. Thus, additional research is needed to better understand the seasonal variations
in excess acidic species around Bermuda.

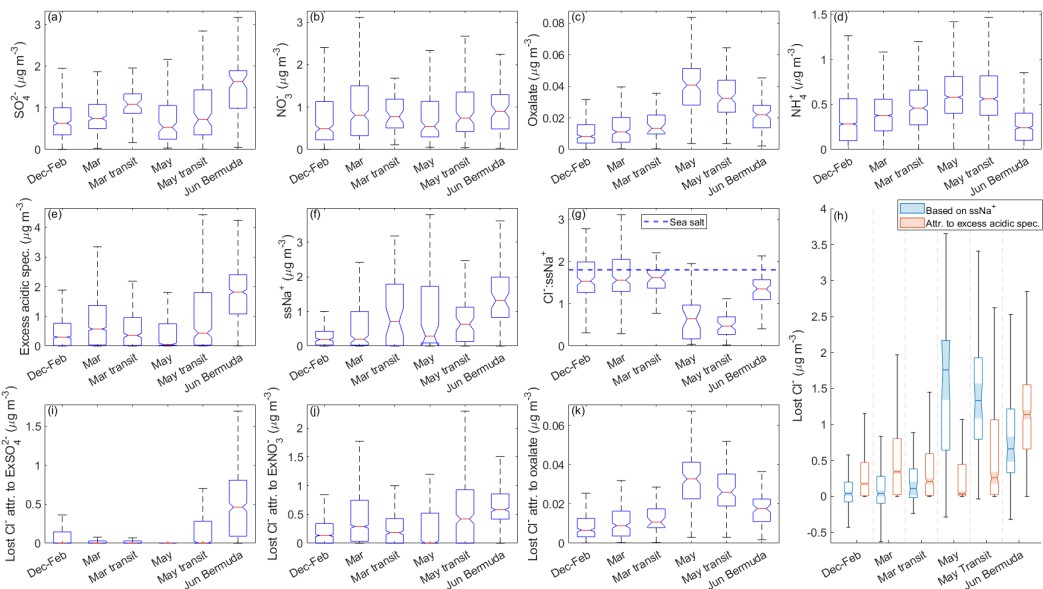


**Figure 4.** Notched box plots showing seasonal/categorical differences in observed mass concentrations from clear ensembles of bulk PILS (**a**) sulfate ($SO_4^{2-}$), (**b**) nitrate ($NO_3^-$), and (**c**) oxalate, as well as (**d**) AMS ammonium ($NH_4^+$). Similar plots are shown for derived mass concentrations of (**e**) total excess acidic species, (**f**) sea salt $Na^+$ ($ssNa^+$), the ratio of (**g**) $Cl^-$:$ssNa^+$, (**h**) mass concentrations of actual and theoretical lost $Cl^-$, as well as theoretical mass concentrations of lost $Cl^-$ attributable to (**i**) excess $SO_4^{2-}$ ($ExSO_4^{2-}$), (**j**) excess $NO_3^-$ ($ExNO_3^-$), and (**k**) oxalate. The value of $Cl^-$:$Na^+$ in sea salt (1.81; Seinfeld and Pandis, 2016) is indicated in **g** with a horizontal dashed blue line. In **h**, light blue boxes represent the actual $Cl^-$ displaced from sea salt particles based on derived mass concentrations of $ssNa^+$, while light red boxes represent the theoretical amount of $Cl^-$ that could have been displaced by the derived mass concentrations of excess acidic species. The properties of the boxes are the same as described in Fig. 1.





### 3.5 Seasonal, spatial, and frontal trends in Cl⁻ depletion

Median ssNa⁺ mass concentrations display similar trends to bulk Na⁺ with comparable values
among the December-February, March, and May categories (0.19, 0.20, and 0.29 µg m⁻³,
respectively), higher mass concentrations for March transit and May transit (0.71 and 0.63 µg m⁻³,
respectively), and highest values for June Bermuda (1.32 µg m⁻³). Median ratios of Cl⁻:ssNa⁺
(1.54, 1.56, 1.62, 0.65, 0.47, and 1.35 for December-February, March, March transit, May, May
transit, and June Bermuda, respectively) are higher than those of Cl⁻:Na⁺ for each category, serving
as a preliminary example of how neglecting contributions of dust to bulk Na⁺ can lead to
overestimates of Cl⁻ depletion. Regardless of magnitude, Cl⁻:Na⁺ and Cl⁻:ssNa⁺ ratios both convey
that the greatest fraction of available sea salt Cl⁻ is converted to reactive chlorine-containing gas
during the month of May (i.e., May and May transit categories) over the NWA. Lost Cl⁻ mass
concentrations are relatively low for December-February, March, and March transit (0.04, 0.04,
and 0.11 µg m⁻³, respectively) then abruptly increase for May and May transit (1.76 and 1.33 µg
m⁻³, respectively) followed by a moderate decrease for June Bermuda (0.66 µg m⁻³). These mass
concentrations correspond to increases in atmospheric mixing ratios of reactive chlorine-
containing gas of 27, 27, 73, 1174, 887, and 440 pptv, respectively, suggesting Cl⁻ depletion
processes have the potential to considerably alter rates of boundary layer VOC oxidation in May
over the NWA; recall that Singh and Kasting [1998] reported ppbv levels of such gases can
produce enough Cl radicals to oxidize 20 – 40% of tropospheric nonmethane alkanes. However,
note our reported lost Cl⁻ mass concentrations are for particles with diameters < 5 µm, so although
May appears to be the only category where Cl⁻ depletion is severe enough to potentially accelerate
tropospheric VOC oxidation, lost Cl⁻ mass concentrations may be higher in reality for other
categories, depending on the extent of depletion reactions in larger sea salt particles.
There is not a clear spatial gradient in lost Cl⁻ over the region (Fig. 5), but mass concentrations
decrease near the USEC after passing frontal systems (Fig. S9), both of which are intuitive as bulk
Na⁺ and excess acidic species mass concentrations display the same trends. Although median lost
Cl⁻ mass concentrations are above 0 for all categories, negative lost Cl⁻ mass concentrations are
observed in 45, 42, 35, 3, 2, and 14% of the samples for December-February, March, March transit,
May, May transit, and June Bermuda, respectively. Negative lost Cl⁻ values can be interpreted as
there being more Cl⁻ in a sample than expected for unreacted sea salt particles based on derived
mass concentrations of ssNa⁺. Such values may indicate influence from non-sea salt sources of Cl⁻
, such as biomass burning (Jing et al., 2017; Park et al., 2013; Cao et al., 2016), mineral dust
(Sullivan et al., 2007), and waste incineration (Moffet et al., 2008). Especially in December-
February and March, negative mass concentrations of lost Cl⁻ often occur in samples with
relatively high mass concentrations of bulk Ca²⁺ (Fig. S10) and K⁺ (Fig. S11), which can be
considered tracers for many of the non-sea salt sources of Cl⁻ mentioned above. However, there
are several exceptions to these relationships, and we leave a more thorough investigation into non-
sea salt sources of particulate Cl⁻ to future studies.



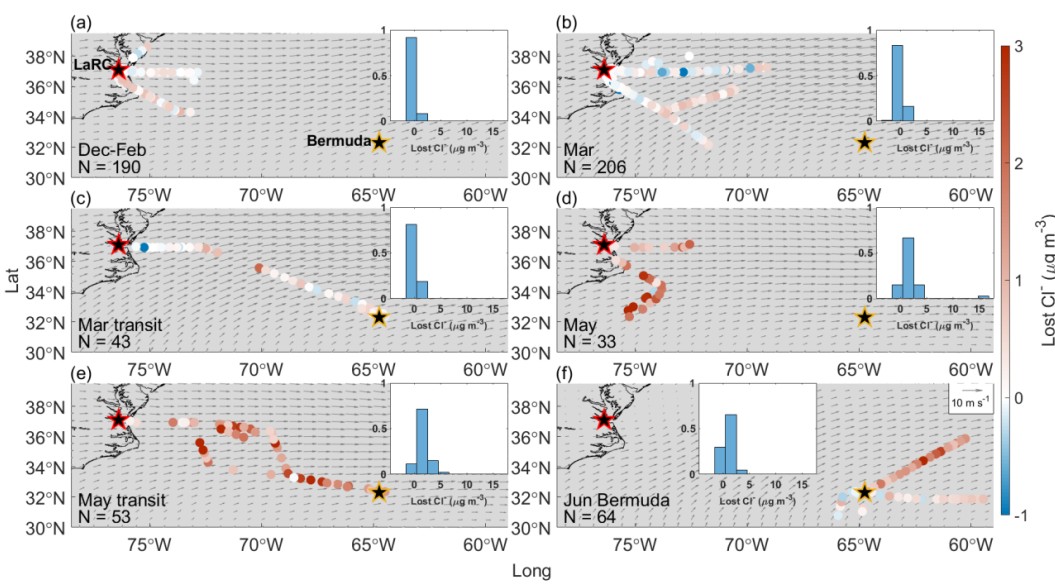


**Figure 5.** Same as Fig. 2, except for lost Cl⁻.





### 3.6 Attributing lost Cl⁻ to acidic species

Median mass concentrations of excess acidic species have the potential to displace 0.17, 0.34, 0.21, 0.04, 0.26, and 1.14 µg m$^{-3}$ (117, 228, 141, 27, 172, and 758 pptv, respectively) of Cl⁻ from sea salt particles for December-February, March, March transit, May, May transit, and June Bermuda, respectively. These hypothetical losses exceed actual mass concentrations of lost Cl⁻ for all categories except May and May transit, suggesting measured excess acidic species often did not react to their full potential with available particulate Cl⁻, considering median %Cl⁻ depletion values are 6, 10, 9, and 64% for December-February, March, March transit, and June Bermuda, respectively. The extent of depletion reactions in December-February, March, March transit, and June Bermuda may have been limited by meteorological variables (e.g., temperature, RH) and/or restricted access of acidic species to particulate Cl⁻ due to the size distribution and/or mixing state of sea salt particles (Su et al., 2022 and references therein).

Most lost Cl⁻ can be attributed mostly to ExNO$_3$⁻ in December-February, March, March transit, and May transit, which is consistent with findings from past works (e.g., Nolte et al., 2008; Yao and Zhang, 2012; Zhao and Gao, 2008). Excess SO$_4$$^{2-}$ and ExNO$_3$⁻ have the potential to contribute equally to Cl⁻ losses for June Bermuda, yet since actual lost Cl⁻ was much lower than theoretical lost Cl⁻, the extent to which each species contributed is unknown. Oxalate has the potential to displace the least Cl⁻ for all categories (0.01, 0.01, 0.01, 0.03, 0.03, and 0.02 µg m$^{-3}$ for December-February, March, March transit, May, May transit, and June Bermuda, respectively), although it is but one organic acid among thousands (Robinson et al., 2007). As mentioned above, there is convincing evidence that organic acids had considerable presence in sampled air masses, especially for Mar transit, May transit and May. This may be due to rising amounts of incident solar radiation accelerating photochemical oxidation of abundant biogenic and anthropogenic VOCs along the USEC to produce secondary organic aerosols (SOA), followed by further oxidation of these SOA to produce oxygenated organics, many of which can serve as weak acids in Cl⁻ depletion reactions. It is possible that unmeasured organic acids are responsible for the lost Cl⁻ that currently cannot be accounted for in May and May transit, although further research is necessary to explore this idea, specifically studies quantifying mass concentrations of additional organic acids in the context of Cl⁻ depletion.

### 3.7 Outcomes from quantifying Cl⁻ depletion semi-unconventionally

In the following subsections we examine the effects of accounting for (i) dust and (ii) dust and combustion emissions as a source of Na⁺, as well as focusing our discussions on mass concentrations of Cl⁻ displaced from sea salt particles instead of either %Cl⁻ depletion or Cl⁻:Na⁺ ratios alone. We consider these to be "semi-unconventional" approaches as a handful of studies have employed at least one of these methods, but they are not commonly used in Cl⁻ depletion studies (based on the 76 studies presented in Table S3 in Su et al., 2022). However, we acknowledge many works neglect non-sea salt sources of Na⁺ after determining crustal contributions are unlikely (e.g., Rastogi et al., 2020; Bondy et al., 2017) or avoid calculating Cl⁻ depletion for particles of a certain size range when anthropogenic sources seem to contribute to Na⁺ and/or Cl⁻ (e.g., Feng et al., 2017; Nolte et al., 2008). This work builds on past studies to provide an all-encompassing method for quantifying Cl⁻ depletion in air masses influenced by dust and/or combustion emissions, as well as relating Cl⁻ losses to their potential effects on atmospheric oxidation processes. We now discuss when, if ever, these methods are of importance for the NWA and provide a few lessons learned for future works interested in using these methods.



### 3.7.1  Significance of accounting for $Na^+$ in dust

To facilitate understanding of the results below, recall mass concentrations of lost $Cl^-_{diff}$ quantify the difference in estimating $Cl^-$ depletion when dust is considered as a source of $Na^+$ (Approach 1) versus when $Na^+$ is attributed entirely to sea salt (Approach 2). Median lost $Cl^-_{diff}$ mass concentrations are 0.05, 0.1, 0.09, 0.05, 0.02, and 0.01 µg m$^{-3}$ (33, 64, 59, 34, 11, and 7 pptv, respectively) for December-February, March, March transit, May, May transit, and June Bermuda, respectively, meaning that $Cl^-$ losses are overestimated by a factor of 2.24, 3.38, 1.80, 1.03, 1.01, and 1.01, respectively, when using Approach 2 versus Approach 1. However, even though overestimates are proportionally large for December-February, March, and March transit, it may not be critical to account for dust as a source of $Na^+$ on a seasonal scale (Fig. 6). Specifically, lost $Cl^-_{bulk}$ mass concentrations for December-February, March, and March transit (58, 91, and 133 pptv, respectively) are still well below the point where they would significantly accelerate VOC oxidation in the boundary layer. Similarly, Approaches 1 and 2 both lead to the conclusion that depletion reactions in May have the potential to accelerate tropospheric VOC oxidation, while lost $Cl^-_{diff}$ values are too small for May transit and June Bermuda to affect overarching conclusions regarding relationships between $Cl^-$ depletion and VOC oxidation rates. However, this study reports mass concentrations of lost $Cl^-$ and lost $Cl^-_{diff}$ for particles with ambient diameters < 5 µm, so it is possible that contributions of $Na^+$ from dust particles > 5 µm may be sufficiently high to lead to critical overestimates in $Cl^-$ depletion, especially considering that lost $Cl^-$ mass concentrations may increase when additionally accounting for depletion in larger sea salt particles.

Although not critically important on a seasonal scale, Approaches 1 and 2 produce considerably different estimates of lost $Cl^-$ for several flights sampling air masses more heavily influenced by dust. Median bulk $Ca^{2+}$ mass concentrations are 5.2 and 8.2 times higher on 30 November and 01 December 2021 (RFs 94 and 95, respectively) than the December-February median without corresponding enhancements in bulk $Na^+$, suggesting a higher presence of dust than usual. Using Approach 1, 100% and 88% (0.14 and 0.23 µg m$^{-3}$, respectively) of median bulk $Na^+$ mass concentrations are attributed to dust for 30 November and 01 December (Table S7), respectively, which results in corrections of lost $Cl^-$ up to 0.63 µg m$^{-3}$ (420 pptv) compared to overestimates based on Approach 2 (Fig. 7). Dust particles sampled on these flights were likely lofted in smoke plumes extending over the NWA from fires in the eastern and southeastern U.S. On 03 March 2022 (RFS 131 and 132), median bulk $Ca^{2+}$ and $Na^+$ mass concentrations are 2.1 and 3.4 times higher, respectively, than categorical medians, as it appears the NWA was heavily influenced by BB emissions from agricultural fires throughout the eastern U.S. Although only 15% of the median bulk $Na^+$ mass concentration is attributed to dust, lost $Cl^-_{diff}$ mass concentrations are as high as 1.05 µg m$^{-3}$ (700 pptv), with most between 0.11 and 0.32 µg m$^{-3}$ (73 - 213 pptv). As mentioned in Sect. 3.3, there is interest in exploring the spatial gradient in bulk $Ca^{2+}$ along March transit flights (RFs 142 and 143) to see how estimates of $Cl^-$ depletion are affected by the transition from a potentially dust-influenced air mass (directly east of LaRC) to one with less dust influence (to the southeast towards Bermuda). Although lost $Cl^-_{diff}$ mass concentrations are lower compared to those of previous case studies, Approach 2 overestimates $Cl^-$ depletion more for the air mass closest to the USEC compared to that closest to Bermuda. The air mass with higher bulk $Ca^{2+}$ mass concentrations appears to be composed of emissions from widespread springtime BB, and the shape of the plume is such over the NWA that the aircraft would fly in it near the USEC but not necessarily near Bermuda. The case studies above suggest that $Cl^-$ depletion can be considerably overestimated in smoke plumes when using Approach 2 as entrained dust particles can contribute meaningfully to bulk $Na^+$ mass concentrations, and that these overestimates may be of





consequence when relating $Cl^-$ depletion to potential increases in VOC oxidation over the region.
Median $Ca^{2+}$ mass concentrations are 3 times higher (0.21 µg m$^{-3}$) than the June Bermuda median
on 11 June 2022 (RFs 172 and 173) without similar enhancements in bulk $Na^+$, suggesting
increases in bulk $Ca^{2+}$ are likely due to African dust sampling (as opposed to increased sea salt
mass concentrations). The arrival of African dust near Bermuda results in overestimates of lost $Cl^-$
up to 0.315 µg m$^{-3}$ (210 pptv) via Approach 2, which are not large enough to affect predictions for
potential increases in rates of tropospheric VOC oxidation. Sampling ended near the beginning of
the peak season for long-range transport of African dust to the NWA (e.g., Prospero, 1996;
Zuidema et al., 2019), so we do not have many flights to choose from for studying effects of
African dust plumes on $Cl^-$ depletion calculations. Using 6300 ppm as a mass ratio of $Na^+$ in dust
particles (Seinfeld and Pandis, 2016), 131.54 and 73.66 µg m$^{-3}$ of dust would be necessary to cause
critical overestimates of lost $Cl^-$ (i.e., lost $Cl^-_{bulk}$ values would reach 1.5 µg m$^{-3}$ using Approach 2)
assuming 0 and 0.66 µg m$^{-3}$ of $Cl^-$ were already being displaced from sea salt particles, respectively
(note 0.66 µg m$^{-3}$ is the median lost $Cl^-$ value for June Bermuda). Edwards et al. (2021) reported
peak African dust mass concentrations of 73.32 µg m$^{-3}$ near Miami, Florida, so it may be possible
for values to reach these levels over Bermuda, but it would take a relatively large plume. Therefore,
it is typically not critical to use Approach 1 when quantifying $Cl^-$ depletion near Bermuda, yet it
may be important to use this approach during strong African dust events.
858        Furthermore, past works have demonstrated the uptake of precursors to acidic species (e.g.,
$NO_x$, $SO_2$; Grassian, 2002; Hanisch and Crowley, 2003; Ullerstam et al., 2002), inorganic acids
(e.g., $H_2SO_4$, $HNO_3$; Ooki and Uematsu, 2005; Sullivan et al., 2007), organic acids (Al-Hosney et
al., 2005; Carlos-Cuellar et al., 2003), and HCl (Zhang and Iwasaka, 2001; Ooki and Uematsu,
2005; Sullivan et al., 2007; Santschi and Rossi, 2006; Sorooshian et al., 2012) on dust particles.
Thus, in addition to considering dust as a source of $Na^+$, it may also be important to account for its
presence to avoid overestimating $Cl^-$ depletion and its impacts on atmospheric oxidation as (i)
uptake of acidic species and their precursors may reduce amounts available for depletion reactions,
and (ii) deposition of HCl on dust particles may reduce the amount of Cl radicals produced
following $Cl^-$ displacement.





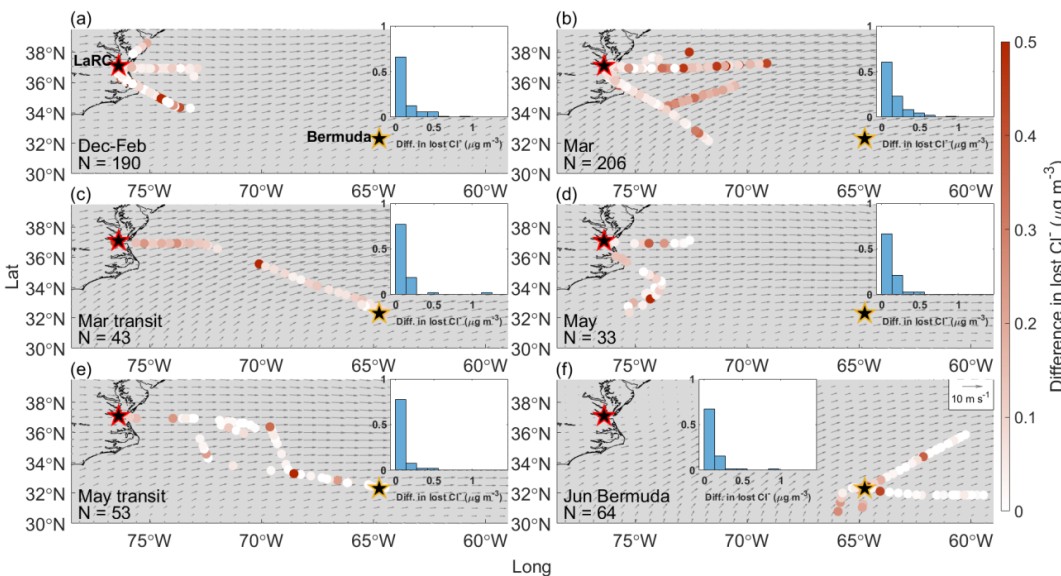


**Figure 6.** Same as Fig. 2, except for differences in lost Cl⁻ when sea salt is assumed to be the only source of bulk Na⁺ versus when sea salt and dust are both considered to contribute to bulk Na⁺ mass concentrations.

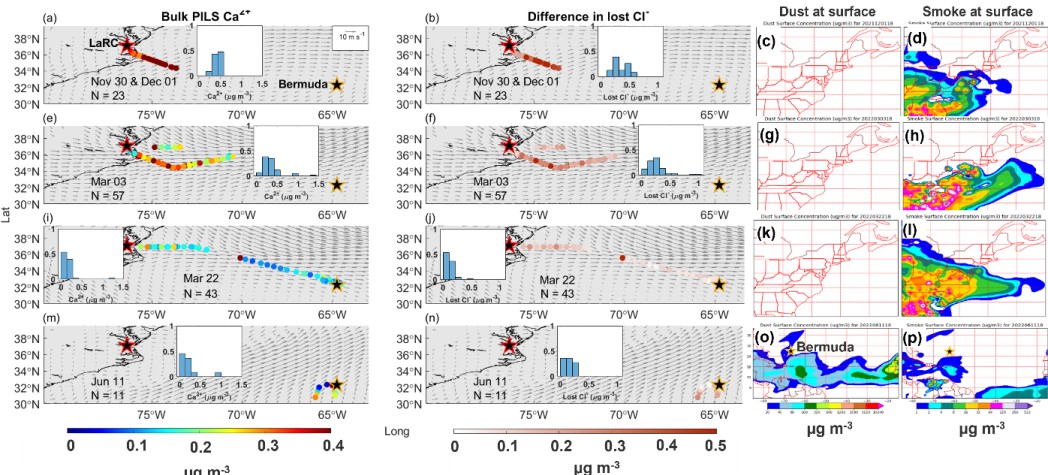


**Figure 7.** Spatial relationships between mass concentrations of (**a**) bulk PILS $Ca^{2+}$ and (**b**) differences in lost $Cl^-$, as well as NAAPS reanalysis surface mass concentrations of (**c**) dust and (**d**) smoke for the case study on 30 November – 01 December 2022 (RFs 94 and 95). The second, third, and fourth rows correspond to case studies on 03 March (RFs 131 and 132), 22 March (RFs 142 and 143), and 11 June (RFs 172 and 173) 2022, respectively, where (**e**, **f**, **g**, **h**), (**i**, **j**, **k**, **l**), and (**m**, **n**, **o**, **p**) display the same variables as (**a**, **b**, **c**, **d**), respectively. Normalized histograms for bulk PILS $Ca^{2+}$ and differences in lost $Cl^-$ show the distribution of values for that specific case study since overlap among the colored dots can hide some from view. Grey arrows indicate the average magnitude and direction of MERRA-2 winds at 950 hPa for the month(s) relevant to each category. NASA Langley Research Center (LaRC) and Bermuda are marked with red-edged and golden-edged stars, respectively.





### 3.7.2 Significance of accounting for $Na^+$ in dust and combustion-sourced particles

As shown above, air masses influenced by BB frequently advect over the NWA, especially in March, occasionally increasing dust mass concentrations to levels capable of causing considerable overestimates in $Cl^-$ depletion. However, there is little to no effect on $Cl^-$ depletion calculations when accounting for contributions to $Na^+$ from combustion particles emitted via agricultural burning and forest fires as median $Na^+_{comb}$ mass concentrations are $0.00 \, \mu g \, m^{-3}$ for all categories (Tables S8 and S9, respectively). Therefore, it may be more important to quantify contributions of dust as opposed to the combustion-sourced particles in smoke plumes over the NWA to avoid overestimates of $Cl^-$ depletion. However, recall median bulk $K^+$ mass concentrations for this study are 2 and 14 times lower than values measured in air masses more heavily influenced by (i) agricultural burning (Kacenelenbogen et al., 2022) and (ii) wildfire smoke (Adachi et al., 2022), respectively. Thus, it is possible quantifying $Na^+_{comb}$ is important for accurate estimates of $Cl^-$ depletion in more concentrated BB plumes, yet we cannot explore this with the flights available and leave such an investigation to future studies. When combustion emissions are attributed to industrial operations, residential wood burning in sauna stoves, car driving, or coal burning at power plants, there is also no influence on $Cl^-$ depletion calculations for any category (i.e., all median $Na^+_{comb}$ values are $0.00 \, \mu g \, m^{-3}$; Tables S10 – S13). Thus, particles generated by the myriad of combustion processes occurring along the eastern U.S. may be too dilute over the NWA to affect calculations of $Cl^-$ depletion not only in air masses reaching Bermuda but also in those much closer to the USEC (e.g., Fig. S12).

Since mass concentrations of $Na^+_{comb}$ are typically negligible, Eqs. 1 – 4 and 6 – 13 should provide the same median mass concentrations of $ssNa^+$ and $Na^+_{dust}$ for each category. However, many samples are excluded when using Eqs. 6 – 13 as their $K^+$ mass concentrations are below IC detection limits, causing adjustments in median $ssNa^+$ and $Na^+_{dust}$ values for several categories. Despite the advantages in accounting for non-sea salt sources of $Na^+$, one disadvantage is potential dataset reduction. For example, 275, 246, 48, 113, 106, and 81 samples provide bulk $Na^+$ mass concentrations for December-February, March, March transit, May, May transit, and June Bermuda, respectively, yet only 202, 220, 48, 64, 75, and 66, respectively, can be used in Eqs. 1 – 4, with even fewer available for use in Eqs. 6 – 13 where concurrent mass concentrations of bulk $Na^+$, $Ca^{2+}$, and $K^+$ are necessary. Thus, future studies may want to weigh the consequences of neglecting contributions of $Na^+$ from non-sea salt sources versus potential reductions to the number of samples included in statistical analyses.

### 3.7.3 Significance of focusing on lost $Cl^-$ instead of %$Cl^-$ depletion

Values of %$Cl^-$ depletion display similar trends to lost $Cl^-$ mass concentrations, where most percentages are (i) relatively low for December-February, March, and March transit (half are $\leq$ 10%, nearly all are $\leq$ 50% ), (ii) relatively high for May and May transit (nearly all are > 40%), and (iii) relatively moderate for June Bermuda (values are distributed fairly evenly from 0 – 30%, and nearly all are are $\leq$ 60%, Fig. S13). However, these %$Cl^-$ depletion values can only be used to show relative seasonal/categorical differences, and they cannot (i) inform when $Cl^-$ mass transfer is greatest from the particulate to gas phase or (ii) place such depletion reactions in the context of their potential influence on tropospheric VOC oxidation rates. For example, samples with higher %$Cl^-$ depletion values can easily be misinterpreted as having greater $Cl^-$ losses when in reality the



opposite may be true. Lost $Cl^-$ and %$Cl^-$ depletion have a negative correlation for May and May transit, meaning that samples with the least (most) displaced $Cl^-$ have the highest (lowest) %$Cl^-$ depletion values (Fig. 8). The remaining categories have mostly positive correlations between lost $Cl^-$ and %$Cl^-$ depletion, although %$Cl^-$ depletion values are typically higher for samples with relatively low buk $Na^+$ mass concentrations at a fixed lost $Cl^-$ value. This trend may be due to samples contianing sea salt particles with varying size distributions (i.e., lower bulk $Na^+$ mass concentrations may mean smaller sea salt particles were collected in a given sample) and considering that smaller sea salt particles are typically more susceptible to depletion reactions (e.g., Su et al., 2022 and references therein). However, this behavior may also be an artifact of increased sensitivity of %$Cl^-$ depletion to samples with relatively small $ssNa^+$ mass concentrations. Many of the samples with mass concentrations of lost $Cl^-$ high enough to potentially influence VOC oxidation rates (i.e., lost $Cl^- > 1.5$ µg m$^{-3}$) have %$Cl^-$ depletion values $< 40\%$, while nearly all samples with % $Cl^-$ depletion values $> 80\%$ do not have $Cl^-$ losses capable of affecting such rates. Thus, we highly recommend future studies quantify mass concentrations of lost $Cl^-$ to make results from depletion studies more suitable for understanding mass exchange between sea salt particles and the surrounding atmosphere and the consequences this can have on rates of tropospheric chemistry and radiative forcing.

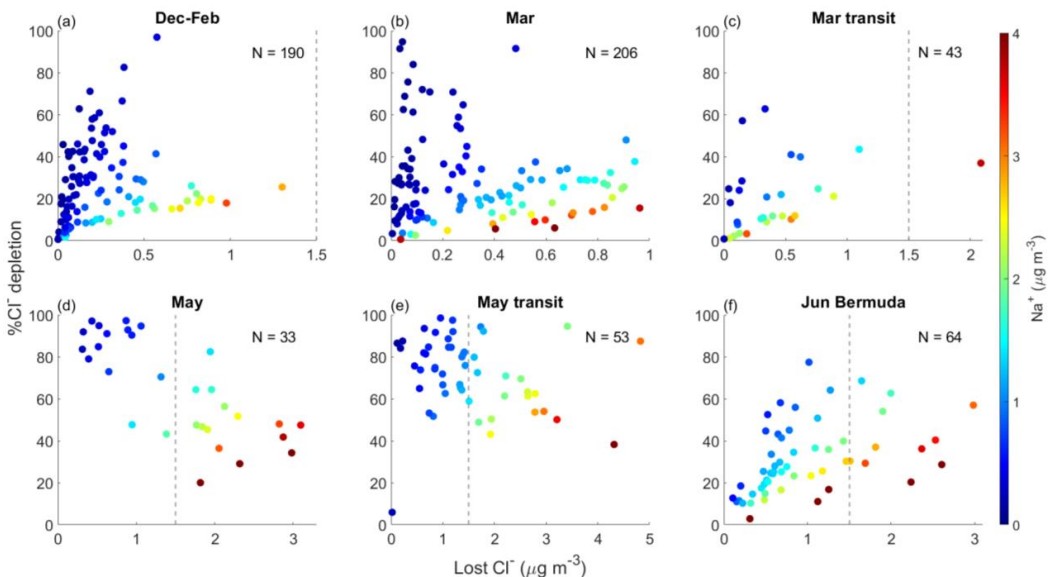

**Figure 8.** Relationships between mass concentrations of lost Cl⁻ and %Cl⁻ depletion for (**a**) December-February, (**b**) March, (**c**) March transit, (**d**) May, (**e**) May transit, and (**f**) June Bermuda. Markers are colored by bulk PILS Na⁺ mass concentrations, and the vertical dashed gray line in some panels denotes where mass concentrations of lost Cl⁻ may begin to have considerable influence on tropospheric VOC oxidation rates.





### 4. Conclusions

This study investigates Cl⁻ depletion in sea salt particles over the NWA from approximately December 2021 – June 2022 using an airborne dataset quantifying the chemical composition of particles < 5 µm among other parameters throughout the lower 3 km of the atmosphere. Trends in bulk PILS $Na^+$ suggest sea salt mass concentrations (1) do not exhibit seasonal variation but are reduced following the passage of MLCs near the USEC, and (2) are higher in the open-ocean environment of Bermuda than along the USEC. Losses of Cl⁻ are greatest in May and least in December-February and March, with median lost Cl⁻ mass concentrations of 1.76, 0.04, and 0.04 µg m⁻³ (1174, 27, and 27 pptv), respectively. Mass concentrations of measured excess acidic species can account for all the Cl⁻ depletion observed in December-February, March, and June near Bermuda, yet none in May, suggesting unmeasured organic acids may be largely responsible for displacement in certain months. Accounting for dust as a source of $Na^+$ is not critical for accurately predicting how Cl⁻ depletion reactions will influence rates of tropospheric VOC oxidation on a seasonal basis, yet this may be important for large smoke and dust plumes over the NWA. Combustion-sourced particles do not contribute enough $Na^+$ to meaningfully affect Cl⁻ depletion estimates in any season for the air masses sampled. Finally, quantifying Cl⁻ depletion as a percentage sufficiently captures seasonal trends in depletion processes but fails to convey the effects they may have on atmospheric oxidation rates.

These results help address several uncertainties regarding Cl⁻ depletion over the NWA and its influence on regional oxidation cycles. First, by identifying factors affecting regional sea salt mass concentrations, we help advance the scientific community towards better understanding and forecasting of regional fluctuations in this major reactive atmospheric Cl reservoir. Additionally, seasonally resolved mass concentrations of lost Cl⁻ reveal that depletion reactions correspond to increases in reactive chlorine-containing gases capable of producing concentrations of Cl radicals sufficient to oxidize 20 – 40% of nonmethane alkanes in the marine troposphere in May, which can have numerous implications including potentially accelerating $O_3$ production over this highly populated region. The possibility for dust to cause meaningful overestimates of Cl⁻ depletion is a regionally novel finding and should encourage future studies and modeling efforts to monitor and account for smoke and dust plumes advecting over the NWA when quantifying sea salt reactivity. Additionally, our results reveal the importance in quantifying absolute Cl⁻ losses as samples with the highest values of %Cl⁻ depletion often have relatively low Cl⁻ losses, and lost Cl⁻ and %Cl⁻ depletion are negatively correlated in May, which is critical to recognize as Cl⁻ depletion has the greatest potential effect on tropospheric VOC oxidation rates during this month compared to all other studied.

Lost Cl⁻ mass concentrations are similar between median values reported in this study and the mean presented in Keene et al. (1990) for summertime conditions around Bermuda (0.66 and 0.68 µg m⁻³, respectively), while our values also fall within the range observed over Bermuda in spring (0.22 – 1.35 µg m⁻³; Keene and Savoie, 1998). Keene et al. (1990) reported lower lost Cl⁻ mass concentrations along the USEC from July-September than our findings in May (1.11 and 1.76 µg m⁻³, respectively), while our median in May is above the range shared in Keene et al. (2007) for July-August (0 – 1.31 µg m⁻³). Haskins et al. (2018) quantified median lost Cl⁻ mass concentrations of 0.30 µg m⁻³ over the ocean from February – March, which is 7 times higher than our medians for December-February and March (0.04 and 0.04 µg m⁻³, respectively), yet note their study specifically targeted polluted winter air masses while ours did not. Many past works along the North American east coast have been able to attribute Cl⁻ depletion largely to inorganic acids in the summer and fall (Zhao and Gao, 2008; Keene et al., 2007; Nolte et al., 2008; Yao and Zhang,



2012), with Keene et al. (1990) reporting a lowest contribution of 38%. We can attribute all Cl⁻ depletion to inorganic acids in December-February, March, and June, yet find inorganic acids do not contribute at all to displacement reactions in May. Our study suggests depletion reactions are still occurring to the extent they were in the 1990s and 2000s over the NWA except that organic acids are possibly becoming increasingly responsible for Cl⁻ displacement, especially in May, although further research is needed to verify this.

Although the ACTIVATE dataset is well-equipped to explore seasonal and spatial trends in Cl⁻ depletion over the NWA, there are several caveats and limitations to be mindful of when reviewing our results. Reported mass concentrations of sea salt and lost Cl⁻ should be interpreted as a lower limit due to the size range of particles sampled ($< 5$ µm). Additionally, calculations for the neutralization of $SO_4^{2-}$ and $NO_3^-$ by $NH_4^+$ combine speciated mass concentrations from two separate instruments, each considering a different size range of particles, meaning mass concentrations of excess acidic species should be considered as an upper limit for particles $< 5$ µm. We recommend accounting for non-sea salt sources of $Na^+$ when appropriate but acknowledge that it may limit statistical analyses as the procedure for disentangling contributions of various sources to bulk $Na^+$ requires synchronous mass concentrations of multiple species.

Overall, this study presents an updated account of sea salt reactivity over the NWA while also providing unprecedented statistics for (i) responses in parameters relevant to Cl⁻ depletion to passing frontal systems, (ii) sea salt particle mass concentrations within the lower 3 km of the atmosphere between the USEC and Bermuda, (iii) the extent of Cl⁻ depletion occurring in a variety of air masses in winter, spring, and early summer as well as the importance of (iv) accounting for smoke and dust plumes as a source of $Na^+$ and (v) quantifying Cl⁻ depletion absolutely instead of relatively. Our finding that depletion reactions are extensive enough to alter rates of VOC oxidation along the USEC in May is impactful on multiple levels ranging from human health to regional radiative forcing, while reporting that inorganic acidic species are not contributing to these losses informs future works and the chemical modeling community that additional acidic species are critical to first identify and then to monitor. Finally, this study reveals the limitations in using traditional methods when quantifying Cl⁻ depletion and will hopefully motivate future works to either be mindful of these limitations or choose alternative methods.

**Data availability**

The          ACTIVATE          dataset          can          be          found          at https://doi.org/10.5067/SUBORBITAL/ACTIVATE/DATA001 (ACTIVATE Science Team, 2020). Level-3 (8-day, 4 km resolution) sea surface chlorophyll a concentrations from MODIS-Aqua can be found at https://doi.org/10.5067/AQUA/MODIS/L3M/CHL/2022.

**Author contributions**

YC, ECC, JPD, GSD, CER, MAS, ELW, and LDZ collected and/or prepared the data. ELE conduction the data analysis. ELE, ECC, and AS conducted data interpretation. ELE and AS prepared the manuscript with editing from ECC, JPD, GSD, MAS, ELW, and LDZ.





**Competing interests**

At least one of the (co-)authors is a member of the editorial board of Atmospheric Chemistry and Physics.

**Disclaimer**

Publisher's note: Copernicus Publications remains neutral with regard to jurisdictional claims in published maps and institutional affiliations.

**Acknowledgements**
The authors acknowledge Claire Robinson for her contributions to this study and dedicate this to her. We thank pilots and aircraft maintenance personnel of NASA Langley Research Services Directorate for successfully conducting ACTIVATE flights and all others who were involved in executing the ACTIVATE campaign.

**Financial support**

This work was funded by ACTIVATE, a NASA Earth Venture Suborbital-3 (EVS-3) investigation funded by NASA's Earth Science Division and managed through the Earth System Science Pathfinder Program Office. University of Arizona investigators were funded by NASA grant no. 80NSSC19K0442 and ONR grant no. N00014-21-1-2115.



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
