# Peer review of "Sea salt reactivity over the northwest Atlantic: An in-depth look using the airborne ACTIVATE dataset"

_EGUsphere, 2023_

## Author Response (AR1)

Eva-Lou Edwards

Response file to comments on "Sea salt reactivity over the northwest Atlantic: An in-depth look using the airborne ACTIVATE dataset"

https://doi.org/10.5194/egusphere-2023-2575

**Reviewer 1**

This manuscript makes use of an extensive ACTIVATE dataset to reassess chloride depletion from aerosol particles. The authors use a series of equations and literature-based ratios for ions of sea salt, dust and emissions from various combustion processes to derive chloride depletion. The goal of this study is to produce and updated, multi-seasonal and geographically expanded account of sea salt reactivity over the NWA. This is a difficult task given the rapid fluctuations in synoptic-scale weather during field campaigns that covered different regions. But, given those difficulties, this is a well documented discussion of the data obtained. Previous studies are also well documented and the manuscript and supplementary data includes numerous tables summarizing the data. The manuscript is well-written and my comments are minimal.

1. The PILS was operated without denuders. AMS NH4+ data were used instead of the PILS NH4+ since the PILS data included gas phase NH3. What about the SO2 that would also be collected in the PILS?

Response: Great question. We have added the following text on Lines 265 – 275 to address this:

"The PILS was operated without upstream acid and base denuders since (1) the removal efficiency for specific relevant gases is not well quantified, (2) it is not known how the removal of gases affects the particle-phase equilibrium for semi-volatile species (e.g., $NO_3^-$), and (3) the addition of denuders decreases the transmission efficiency of coarse-mode sea salt particles into the PILS. While there could be a small positive artifact from certain gases (e.g., $SO_2$, $HNO_3$), the PILS should be much less sensitive to this issue than filter collection methods with offline analysis. However, the absence of a base denuder opened the possibility for $NH_3$, a highly soluble trace gas, to contribute to particulate $NH_4^+$ mass concentrations. During quality control analyses, PILS $NH_4^+$ mass concentrations were unjustifiably high in many samples, prompting us to omit this species from this study's analysis."

2. The PILS and AMS suffer from collection efficiencies less than 100%. How were these instruments corrected for collection losses?

Response: Great question. There were no corrections for collection losses as AMS and PILS sulfate mass concentrations agreed well throughout the campaign. If there were a very obvious, significant, and consistent offset between the instruments, we would have considered reducing AMS collection efficiency (CE), but that evidence didn't seem to exist. There are many factors that contribute to variability in the AMS CE (bulk composition, ambient RH, particle mixing

state, particle morphology) that cannot realistically be compensated for, so changing the CE dynamically (case-by-case) would likely introduce more uncertainty.

To succinctly address this, we have added the following sentence to the manuscript on Lines 281 – 283:

"The AMS collection efficiency was set to unity as there was not compelling evidence to lower this value when comparing AMS and PILS $SO_4^{2-}$ mass concentrations."

**Reviewer 2**

This work gives a highly detailed and thorough analysis of the potential for sea-salt chloride to be displaced from the particle phase as a chlorine containing gas phase compound over the northwest Atlantic by analyzing trends from the ACTIVATE aircraft campaign. Overall, this paper uniquely provides much needed depth and insightful discussion when compared to the existing literature on the subject on both (1) how Cl- depletion is calculated and how important the consideration of sources of non-sea-salt Na+ from other sources may be in deriving these values from observations, as well as (2) how Cl- depletion varies seasonally, spatially, and meteorologically. The only major comment I had that I believe necessitates minor revision is that the authors lean heavily in the text on the potential of the mass of Cl- that is displaced to contribute to tropospheric VOC oxidation to qualify the significance of their findings on line**(s) 56, 61-62, 167-168, 477-478, 725-726, 866-867, 973-974**. Below I've suggested some alternative and more updated values the authors may consider citing that more accurately give lower #s of the potential significance of chlorine radicals on VOC oxidation. However, overall, given the significance of these contributions to the existing literature, I found the paper to be of high scientific quality and value worthy of publication in ACP, but will also note the work is particularly well written and easily digestible, even though I found the discussion to be a bit long winded in some places.

**Specific Comments:**

**Line(s) 56, 61-62, 167-168, 477-478, 725-726, 866-867, 973-974**. There is little discussion of the fact that much of the displaced Cl- in the non-polluted air masses they intersect is likely to be displaced as HCl, a rather inert chlorine containing gas, subject primarily to deposition processes, and is not likely to result in the production of Cl radicals like the displacement that particulate Cl- as Cl2, HOCl, ClNO2, ClNO3, BrCl, or IC would. In particular, I found that the authors cite some outdated literature on the potential impact of Cl radicals on tropospheric VOC oxidation and note that several other studies have more recently attempted to observationally constrain these values and model exactly how much chloride is displaced into different chlorine containing gases, which show lower values for the significance of Cl radicals than is quoted. I think it's important to acknowledge all of the past studies that the authors currently cite, but would certainly encourage the authors to include some of the major results from more recent chlorine budget studies below in their discussion in the sections below & those listed above where this same motivation introduced here is referenced later in the paper:

1. **Line 56**: The authors cite Sherwen et al., 2016 to say that Cl radicals may be responsible for 15-27% of VOC oxidation in the global troposphere. This work was the first to include *some* tropospheric halogen chemistry in the global chemical transport model, GEOS-Chem. However, it was merely the first of a series of improvements to the tropospheric halogen mechanism in GEOS-Chem, which primarily focused on the addition of Iodine/bromine chemistry and importantly for this work, did NOT include the displacement of HCl from sea-salt chloride, nor did it include ClNO2 production as there were few observations available at that time to constrain that chemistry. The results presented in Sherwen et al., 2016 was later revised, and more tropospheric halogen chemistry was added in Sherwen et al., 2017, Sherwen et al., 2018, Wang et al., 2019 (which first included well constrained ClNO2 production and serious improvements to

the reaction rates that influence HOCl and Cl2 production from heterogeneous reactions on sea-salt Cl-) and most recently, by Wang et al., 2021 (which was the first to include HCl displacement from sea-salt Cl-). Therefore, the series of publications on the impacts of halogen chemistry simulated in GEOS-Chem prior to Wang et al., 2021 do not include a full or accurate accounting of the impact of Cl radicals on the tropospheric oxidant budget. As such, I would encourage the authors to update the citation in this line to Wang et al., 2021 and instead cite those #s for the impact of Cl radicals on the tropospheric VOC oxidation budget as the values given in the partial revisions of the impact of tropospheric chlorine chemistry on atmospheric oxidation processes have changed significantly between those series of publications and likely reflect a far more accurate value than those cited in Sherwen et al., 2016.

Response: We deeply thank the reviewer for pointing this out and leading us to more recent literature.

We have updated our statistics regarding the effect of Cl radicals on atmospheric organic oxidation rates. Lines 55 – 59 now read: "Cl radicals may be responsible for 0.8% of the global oxidation of methane, 14% of ethane, 8% for propane, and 7% for longer-chain alkanes (Wang et al., 2021) and can play an exceptionally critical role in governing atmospheric composition in the early morning when OH radicals are less abundant (Young et al., 2013; Riedel et al., 2014; Osthoff et al., 2008)."

Additionally, we added a line to address the reviewer's comment above that stated, "There is little discussion of the fact that much of the displaced Cl- in the non-polluted air masses they intersect is likely to be displaced as HCl, a rather inert chlorine containing gas, subject primarily to deposition processes, and is not likely to result in the production of Cl radicals like the displacement that particulate Cl- as Cl2, HOCl, ClNO2, ClNO3, BrCl, or IC would."

Lines 76 – 79 read: "Note that most of the generated HCl is removed by deposition, but a fraction (~16% globally; Wang et al., 2021) reacts with OH to produce Cl radicals, which initiates rapid cycling between these radicals and their inorganic non-radical reservoirs."

2. **Line 61-62**: The authors (appropriately) cite Keene et al., 1999 to give approximations of the total mass of reactive chloride in the troposphere. However, the values cited in Wang et al., 2021, are different and include a much more updated and better constrained accounting of the tropospheric Chlorine budget. I would likewise encourage the authors to add a line or two discussing the differences between the values between Keene et al., 1999 and Wang et al., 2021 in this section.

Response: Thank you for exposing us to this updated reference. We are happy to discuss the differences between the referenced values.

Lines 62 – 68 now read: "Sea salt aerosol particles are the largest reservoir of reactive atmospheric Cl. Keene et al. (1999) estimated that at any given time there are ~22 Tg of reactive Cl in the troposphere, and that 68% of this mass is found in particulate form, primarily sea salt. More recently, Wang et al. (2021) suggested there are 2.44 Tg of reactive tropospheric Cl with

90% in particulate form as sea salt. The fact that estimates for the reactive tropospheric Cl budget have decreased by an order of magnitude over the past two decades motivates continued research on tropospheric halogen chemistry and its impacts."

Additionally, Singh and Kasting (1988) specifically reported that ppbv concentrations of HCl had the potential to produce sufficient Cl radicals to oxidize 20 – 40% of nonmethane alkanes in the marine troposphere. We have changed the following lines to be more specific and reflective of these findings by replacing phrases resembling "reactive chlorine-containing gases" with "HCl." These changes were made on lines 175, 742, and 990.

**Section 2.4:** I found the thorough discussion of the methodology in this section to use other ratios of ions as markers of biomass burning, dust, and organic acids beyond just that of sea-salt to more accurately calculate the Cl- depletion of high scientific value that will certainly be employed in future studies and will likely have the effect of making this a highly cited work. I note, primarily for the editor, that the equations provided in this section are of key importance for the scientific value of the paper and would strongly suggest that they remain in the main text of the paper and NOT be moved to the supplemental materials if there are other review comments about the overall length of the paper.

Response: Thank you for the feedback. We hope the editor chooses to keep these equations in the main text.